# A droplet robotic system enabled by electret-induced polarization on droplet

Ruotong Zhang [1], Chengzhi Zhang[1,2], Xiaoxue Fan[1], Christina C. K. Au Yeung[1,3], Huiyanchen Li[3], Haisong Lin [1,3] ✉ & Ho Cheung Shum [1,3] ✉

Robotics for scientific research are evolving from grasping macro-scale solid materials to directly actuating micro-scale liquid samples. However, current liquid actuation mechanisms often restrict operable liquid types or compromise the activity of biochemical samples by introducing interfering mediums. Here, we propose a robotic liquid handling system enabled by a novel droplet actuation mechanism, termed electret-induced polarization on droplet (EPD). EPD enables all-liquid actuation in principle and experimentally exhibits generality for actuating various inorganic/organic liquids with relative permittivity ranging from 2.25 to 84.2 and volume from 500 nL to 1 mL. Moreover, EPD is capable of actuating various biochemical samples without compromising their activities, including various body fluids, living cells, and proteins. A robotic system is also coupled with the EPD mechanism to enable full automation. EPD's high adaptability with liquid types and biochemical samples thus promotes the automation of liquid-based scientific experiments across multiple disciplines.

Robotic systems begin to revolutionize scientific experiments from labor-intensive empirical practices to automated standardized operations. Due to the high efficiency, precision, and reliability[1–3], robots have been applied to accelerate drug discovery[4–6], synthesize precision materials and structures[7–9], and facilitate medical diagnostics[10–12]. For these applications in scientific research, automated liquid manipulation is one of the imperative functions since samples and reagents in liquid format are often involved. Automated liquid manipulation can help to increase efficiency with limited labor, while also minimizing manual errors during experimental process[13,14].

Current automation of liquid manipulation mainly requires liquid handling workstations, which transfer liquids by controlling experimental consumables, such as pipettes, centrifuge tubes, microplates, etc[15–17]. However, such systems typically require a prohibitive cost, usually tens of thousands of dollars to >$100,000[18], as well as additional laboratory environment and resources to support[7,8]. Moreover, commercial systems are typically non-modular and non-open source[19–21], further constraining their scalability and widespread

adoption in labs. To overcome these challenges, droplet actuation systems for directly manipulating discrete fluids have been proposed[12,22,23]. Compared with bulky robotic arms, droplet actuation systems can process extremely small volumes of liquids and actuate multiple droplets in parallel through spatially crossover manipulation[11,24–27]. In addition, due to its small footprint and flexible operation, it can be easily integrated into other laboratory automation systems[28–30].

The existing common mechanisms for automated droplet actuation include electrowetting-on-dielectric (EWOD, including air-based EWOD and oil-based EWOD), magnetic, acoustic, and thermal-based (Supplementary Fig. 1)[31–37]. However, platforms based on these mechanisms often lack compatibility with the full range of liquid types that form various biochemical samples. For example, EWOD is not suitable for manipulating liquids with low permittivity and poor conductivity since electrowetting relies on the surface charge generated at the solid-liquid interface[38–40]. Besides, EWOD may cause protein adsorption due to the electrostatic interaction with charged

[1]Department of Mechanical Engineering, The University of Hong Kong, Hong Kong SAR, China. [2]Department of Materials Science and Engineering, Southern University of Science and Technology, Shenzhen, Guangdong, China. [3]Advanced Biomedical Instrumentation Centre, Hong Kong Science Park, Hong Kong SAR, China. ✉e-mail: linhs@hku.hk; ashum@hku.hk

electrodes and the hydrophobic interaction with substrates, leading to biofouling[41–43]. As for magnetic-based systems, the introduced magnetic nanoparticles may catalyze the reaction of certain reagents due to their intrinsic peroxidase-like activity[44,45]. The non-transparent particles could also interfere with the optical properties of droplets, resulting in inaccurate test results[46]. Moreover, both surface acoustic waves and thermal-based methods require externally applied high energy. The increased amplitude of acoustic waves can break cell membranes[47,48], while the generated heat will promote the evaporation of droplets and compromise the activity of DNA/protein/cells[49,50]. Therefore, the existing common systems still lack (1) generality with operable liquid types and (2) compatibility with biochemical samples.

To bridge this technological gap and accommodate a significantly wider range of liquids and biochemical samples, here, we introduce a droplet robotic system based on the observed attraction effect between various liquid droplets and electrets. Electret is a piece of dielectric material that carries a net, macroscopic, quasi-permanent electrostatic charge[51,52]. With the intrinsic electrostatic charges, electret generates a non-uniform electrostatic field, polarizing and attracting droplets, as shown in Fig. 1A. Distinguished from the traditional liquid polarization generated by AC/DC electric field[39,53–56] (Supplementary Note 1), this mechanism of electrostatic charges-generated liquid polarization is termed electret-induced polarization on droplet (EPD). A droplet robotic system is also coupled with the EPD mechanism to fully automate the droplet manipulation process (Fig. 1B), consisting of (1) a programmable control matrix that can generate a regionalized electromagnetic field; (2) EPD grippers made of a hybrid of electret and magnetically responsive materials; (3) target droplet sample. By programming the control matrix, the EPD gripper can actuate droplets in a specific path, such as the shapes of the letter H, K, and U (Fig. 1C, Supplementary Movie 1).

Compared with the existing droplet actuation system, our system shows significant advantages, especially in terms of the range of operable liquid types and compatibility with bio-samples (Fig. 1D, Supplementary Table 1). The EPD-based droplet robotic system, in principle, enables all-liquid actuation and is experimentally validated for actuating various inorganic/organic liquids, including but not limited to water (inorganic), glycerol (alcohol), and triacetin (ester) (Fig. 1D, Supplementary Fig. 2, Supplementary Movie 2), with relative permittivity ranging from 2.25 to 84.2. In contrast, EWOD, both air-based and oil-based, faces challenges when actuating organic liquids with low permittivity (e.g., triacetin with relative permittivity of 7.01) due to the limitation of the electrowetting mechanism. Furthermore, since high-voltage AC/DC electric fields are not required in EPD, the undesired effects of Joule heating and high electric field strength on biological samples can be avoided[57,58]. By validating the actuation of multiple human body fluids, including serum, saliva, and urine (Fig. 1D, Supplementary Fig. 3, Supplementary Movie 3), as well as solutions of protein and living cells, EPD shows superior compatibility with bio-samples compared with EWOD. Besides, EPD also exhibits overwhelmingly collective performances in terms of compatibility with substrates and surroundings (air/oil/air-oil interface), actuation speed

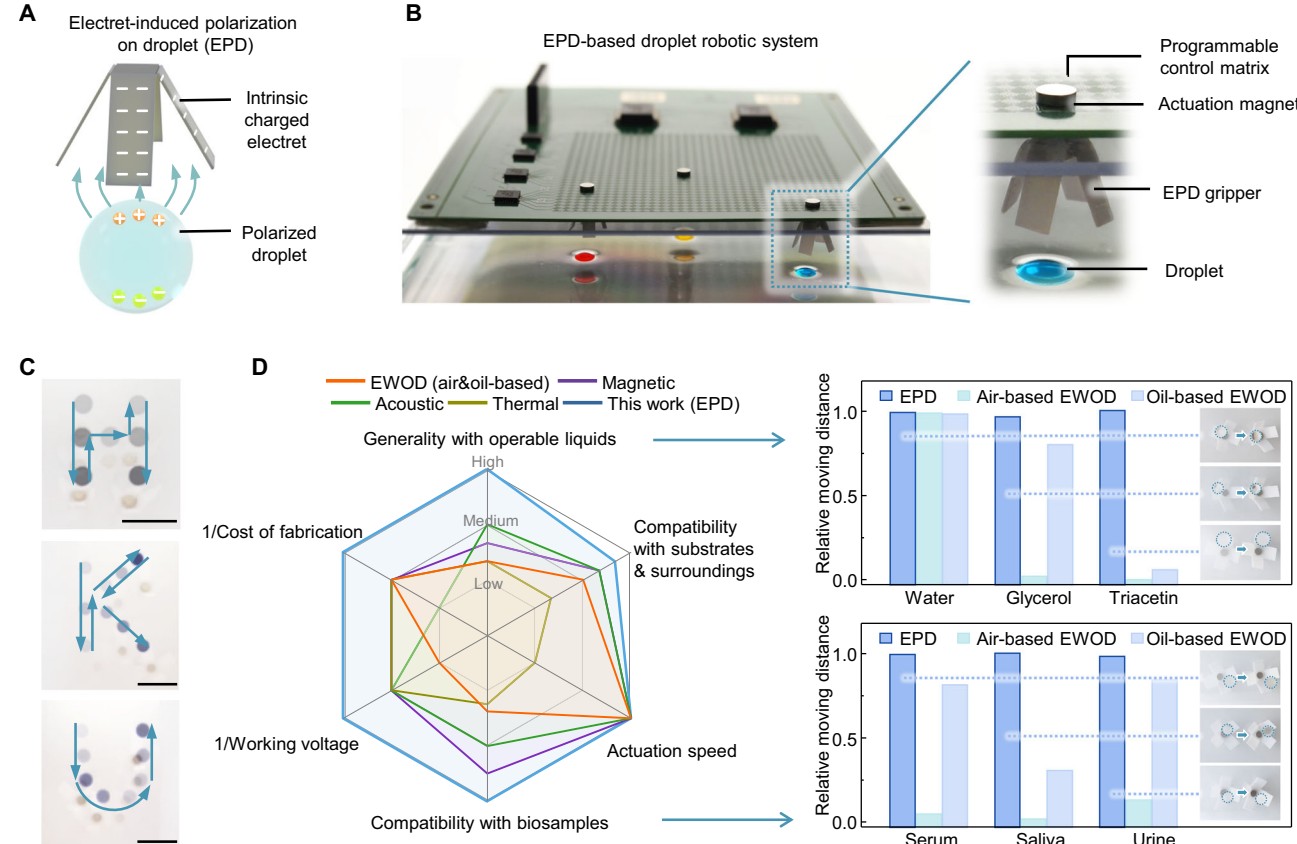

**Fig. 1 | Overview of the droplet robotic system based on the mechanism of electret-induced polarization on droplet (EPD). A** Working principle of the EPD-based droplet actuation. **B** Optical image of the EPD-based droplet robotic system and its core components. **C** Overlaid sequential images (derived from video frames), which visualize the moving path of the droplet actuated by the EPD-based droplet robotic system, including the shapes of letter H, K, and U, respectively. Scale bars: 10 mm. **D** Comparison between the proposed EPD-based droplet robotic system and the existing automated droplet actuation system from 6 perspectives (left). The evaluations are conducted based on the detailed information in Supplementary Table 1. EPD's wide range of operable liquid types and the compatibility with bio-sample are particularly demonstrated and compared with air-based EWOD and oil-based EWOD based on Supplementary Figs. 2, 3, respectively (right). Source data are provided as a Source Data file.

(up to 60 mm/s), working voltage (5.5 V), cost of fabrication (lower than US$1 for consumables and ~US$100 for control system), etc. (Fig. 1D, Supplementary Table 1). These superior properties endow the proposed EPD-based droplet robotic system with the potential to enhance liquid-based scientific experiments in multiple fields, such as clinical, biological, and engineering sciences.

## Results

### Mechanism of EPD and characterization

Our study builds upon a newly observed attraction effect between various liquid droplets and the intrinsically charged electret (Fig. 2A).

When a water droplet is placed at a position ~30 mm away from the negatively charged electret, it is attracted towards the electret with a positive acceleration (Fig. 2B, C) and reverses its direction once it reaches the position below the edge of electret. This attraction between droplet and electret may be explained by two potential hypotheses: (1) droplet net charge-induced movement and (2) droplet polarization-induced movement (Fig. 2D).

In the first hypothesis, due to the contact electrification with other solid surface/immiscible liquids, droplets will tend to obtain electro-static charges of a certain polarity (usually positive for deionized water)[59,60]. Therefore, the electrostatic field formed by the electret may

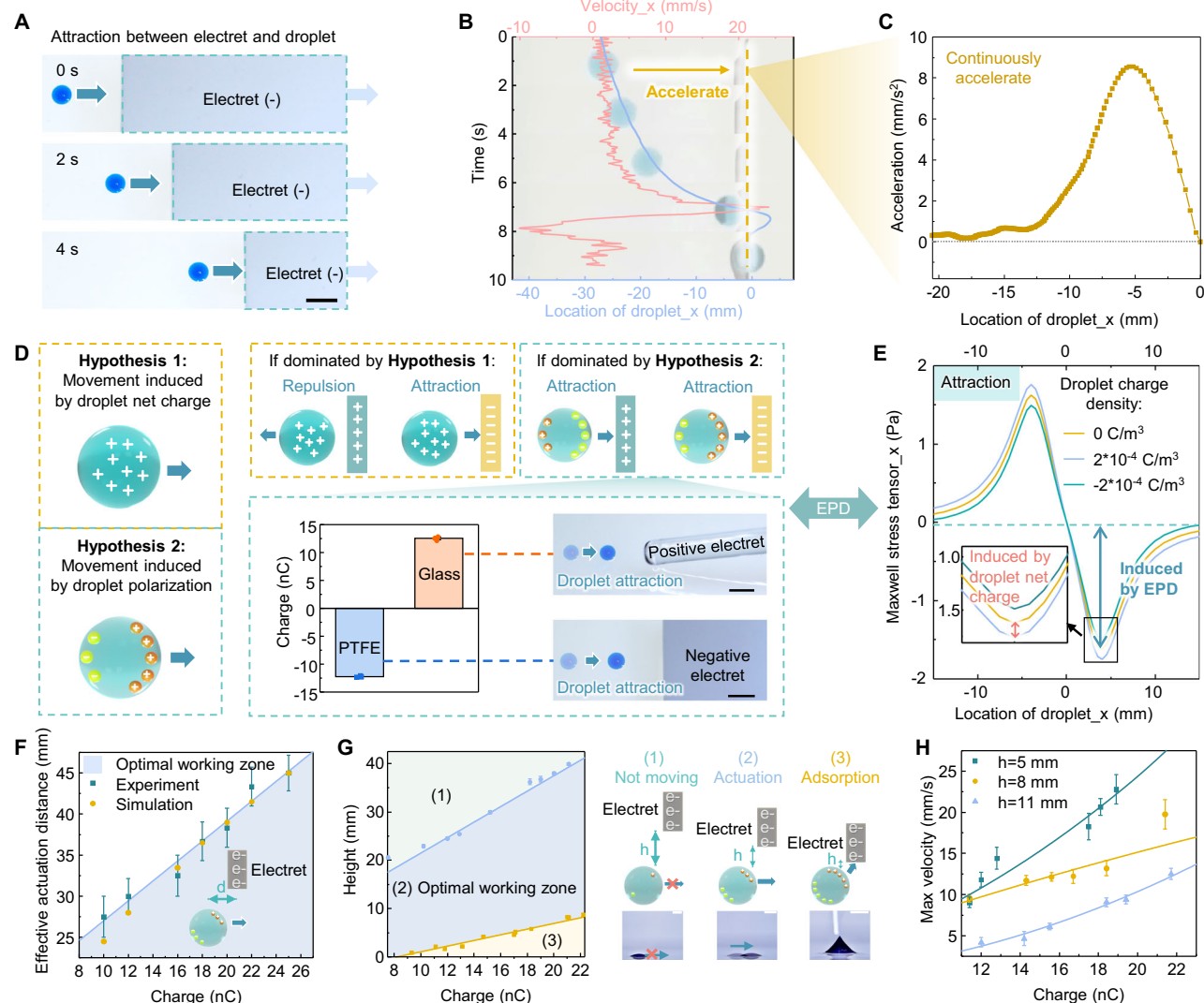

**Fig. 2 | Investigation of the mechanism and characterization for the EPD-based droplet actuation. A** Optical images of the electret-induced attraction of droplet. Scale bars: 5 mm. **B** Motion of a water droplet with the presence of an electret. After oscillations, droplet is immobilized at the edge of the intrinsically charged electret. The blue line represents the position of the droplet over time, and the red line represents the velocity of the droplet. The data are obtained through analyzing the video, and the screenshots of the video at the 1, 3, 5, 7, 9 s are presented in sequence. **C** Acceleration of the droplet during its attraction by a fixed electret. **D** Two potential hypotheses explaining the observed attraction between the electret and droplet (left), while the dominating hypothesis can be determined by actuating droplets with two electrets carrying opposite charge polarities (right). Scale bars: 5 mm. Error bars, SD (n = 3). **E** Simulations of the average Maxwell surface stress tensor applied on droplets with positive, negative, and neutral charge densities, while the set value is determined by the practically measured droplet

charge amount[59]. The results indicate that the force induced by EPD effect (green arrow) is much larger than that induced by droplet net charge (red arrow). **F** The relationship between the effective actuation distance of the droplet and the absolute charge amount of the electret, including both experimental results (blue dot) and simulation results (yellow dot). Error bars, SD. **G** Phase diagram of the droplet dynamic behaviors by adjusting the electret's absolute charge amount and the height difference between the electret and droplet, including (1) droplet staying stationary (green region), (2) normal lateral actuation, which is also the optimal working zone (blue region), and (3) adsorption on the electret (yellow region). The blue line and yellow line represent the maximum and minimum height differences required for effective actuation. Error bars, SD. Scale bars: 2 mm. **H** The relationship between the maximum actuation velocity of a 10 μL droplet and the absolute charge amount of electret under different height differences. Error bars, SD. Source data are provided as a Source Data file.

exert an attractive/repulsive Coulomb force on the charged droplet[61,62]. In the second hypothesis, the electret will generate a non-uniform electrostatic field and polarize droplets into dipoles, termed EPD. Based on the difference in polarizability between the droplets and the surrounding medium, the polarized droplets will move towards/against the direction of the electric field maxima[63,64]. Compared with the traditional AC/DC electric field-induced polarization[65,66], EPD employs the electrostatic charges carried by the electret, eliminating the effect of conduction current and thus leading to different variables and application scenarios (Supplementary Note 1).

In practical situations, force induced by the droplet net charge and droplet polarization may co-exist, which can be described as (Supplementary Note 1)[65,66]:

$$F = \int_{V_{droplet}} \rho E dV + 4\pi\varepsilon_m r^3 \frac{\varepsilon_d - \varepsilon_m}{\varepsilon_d + 2\varepsilon_m}(E \cdot \nabla)E = \int_{V_{droplet}} \int_{S_{electret}} \rho \frac{\varphi dS}{4\pi\varepsilon_m d^2} dV$$
$$+ 4\pi\varepsilon_m r^3 \frac{\varepsilon_d - \varepsilon_m}{\varepsilon_d + 2\varepsilon_m} \left( \int_{S_{electret}} \frac{\varphi dS}{4\pi\varepsilon_m d^2} \cdot \nabla \right) \int_{S_{electret}} \frac{\varphi dS}{4\pi\varepsilon_m d^2} \quad (1)$$

where the first item is induced by droplet net charge, while the second item is induced by polarization, i.e., EPD. $\rho$ is the volumetric charge density of the droplet, $\varphi$ is the surface charge density of the electret, $r$ is the radius of the droplet and is assumed much smaller than the scale of the field nonuniformity[66], $\varepsilon_m$ and $\varepsilon_d$ is the permittivity of the surrounding medium and droplet, respectively, $d$ is the distance between the electret and droplet. To investigate which hypothesis dominates in the droplet actuation process, two electrets carrying opposite charge polarities are utilized to actuate the same water droplet floating on the HFE oil surface (Fig. 2D). Our experimental results in Fig. 2D, consistent with the simulation results (Supplementary Fig. 4), show that water droplets are attracted by both positively charged and negatively charged electret. This result proves that the direction of the generated force $F$ will not change with the polarity of $\varphi$ (Eq. (1)). Therefore, the force induced by EPD is much larger than that induced by droplet net charge, indicating the dominance of EPD effect in our system. The simulated results of the Maxwell stress tensor applied on droplets with positive, negative, and neutral charge densities further support this conclusion that the EPD effect dominates, while the force induced by droplet net charge only accounts for about 8% (Fig. 2E).

To achieve the well-controlled EPD-based droplet actuation, the lateral distance between the electret and droplet should be less than the effective actuation distance, as measured and simulated in the Fig. 2F (Supplementary Fig. 5). Similarly, the vertical distance between the electret and the droplet also needs to fit within an appropriate interval (Fig. 2G). If the height of the electret is too high or the charge density of the electret is too low, the EPD force will be smaller than the lateral resistance, and the droplet will remain stationary. Vice versa, the excessive EPD force will cause the absorption of droplet, resulting in a failed actuation. Within the optimal working state (state 2 in Fig. 2G), the droplet can be actuated horizontally. The maximum actuation velocity can be enhanced by increasing the charge amount of electret or decreasing the height difference between electret and droplet (Fig. 2H, Supplementary Note 2). Based on these empirical relationships, the optimal working zone of the system can be determined, including electret's lateral position, vertical position, carried charge amount, and moving speed, laying a foundation for further automated droplet actuation systems.

### Generality with various operable liquids
In contrast to EWOD's limitations on droplet volume (usually nL-µL, depending on electrode size)[67] and liquid types (usually only for conductive and aqueous liquids)[39], EPD shows advantages in generality with various liquids, including a wider range of droplet volume, liquid conductivity, and liquid permittivity (Fig. 3A). Figure 3B demonstrates the actuation of droplets with volumes ranging from nanoliters to milliliters under EPD effect, including a broad volume range up to four orders of magnitude. As the droplet volume increases, the maximum velocity of droplet movement also increases from 22 mm/s (10 µL) to 60 mm/s (100 µL) (Supplementary Note 3), comparable with other common droplet actuation techniques (Supplementary Table 1). Multiple droplets can also be actuated simultaneously by EPD, as shown in Supplementary Fig. 6.

In terms of liquids with different conductivities, the Maxwell stress tensor generated by EPD is simulated and calculated (Fig. 3C). The results show that the EPD force applied on droplets does not change with conductivity, consistent with the mechanism that EPD uses electrostatic charges to polarize droplet without generating conduction current (Supplementary Note 1). Therefore, for the EPD-based droplet actuation, we can extend the operable range of liquids from conductive to dielectric liquids. For liquids with different relative permittivities, the simulation results show that the force generated by EPD decreases as the permittivity decreases (Fig. 3D). However, the generated EPD force for liquid with the minimum relative permittivity of 1.8 (at normal temperature and pressure[68]) is still much larger than the minimum force required for effective actuation (obtained based on the measured maximum effective actuation distance of water droplet, as shown in the Supplementary Fig. 7). Therefore, our results indicate that EPD has the potential to manipulate all types of liquids.

EPD's adaptability to a wide range of operable liquid types is also experimentally validated. Three common inorganic liquids and three common organic liquids with relative permittivity ranging from 2.25 to 84.2, including alkane, alcohol, and ester, are selected for the experiments (Fig. 3E). The experimental results show that EPD can smoothly actuate various inorganic/organic liquids without changing any parameter setting, since the generated EPD force, even for low permittivity liquid, is strong enough to overcome the moving resistance (Fig. 3D, F). In contrast, EWOD is inadequate for manipulating organic liquids with low permittivity due to the limitation of the electrowetting mechanism, either in the surrounding of air or oil (Supplementary Fig. 2, Supplementary Movie 2). In addition, the simulated effective actuation distances of different droplets can also be obtained from the crossing between the calculated Maxwell stress tensor and the minimum force required for effective actuation (Fig. 3F). The result shows that the effective actuation distance of droplets decreases by up to 26% as the permittivity of droplet decreases, consistent with the experimental results (Fig. 3G). Based on both simulation and experiment results, the EPD effect exhibits a superior generality with a wider range of operable liquids, showing the potential for all-liquid handling.

### Compatibility with diverse biochemical samples and substrates
Considering that droplet robotics are often employed to carry bio-samples[69], here, we also conduct evaluations of EPD's biocompatibility. Our evaluation encompassed a range of bio-samples, including body fluids, proteins, and living cells (Fig. 4A). As for body fluids, human serum, saliva, and urine are tested, and EPD can successfully actuate different body fluids, without modifying any parameter setting (Fig. 4B). However, when manipulating these body fluids on a common double-plate EWOD, droplet actuation becomes challenging, and the actuation performance varies among different body fluids (Supplementary Figs. 3 and 8, Supplementary Movie 3).

The different performance of actuating various body fluids between EPD and EWOD may be attributed to two reasons, one of which is the wide variation in electrical conductivity of different body fluids, affecting the setting of working voltage and actuation robustness of EWOD[70,71]. On the contrary, EPD is proved to be independent of liquid conductivity, as shown in Fig. 3C (Supplementary Note 1). The other reason is that proteins in the body fluids appear to be significantly adsorbed on the substrate of EWOD, especially on the air-based EWOD (Supplementary Figs. 9, 10, Supplementary Movie 4). Electrostatic interactions with the charged electrode and hydrophobic

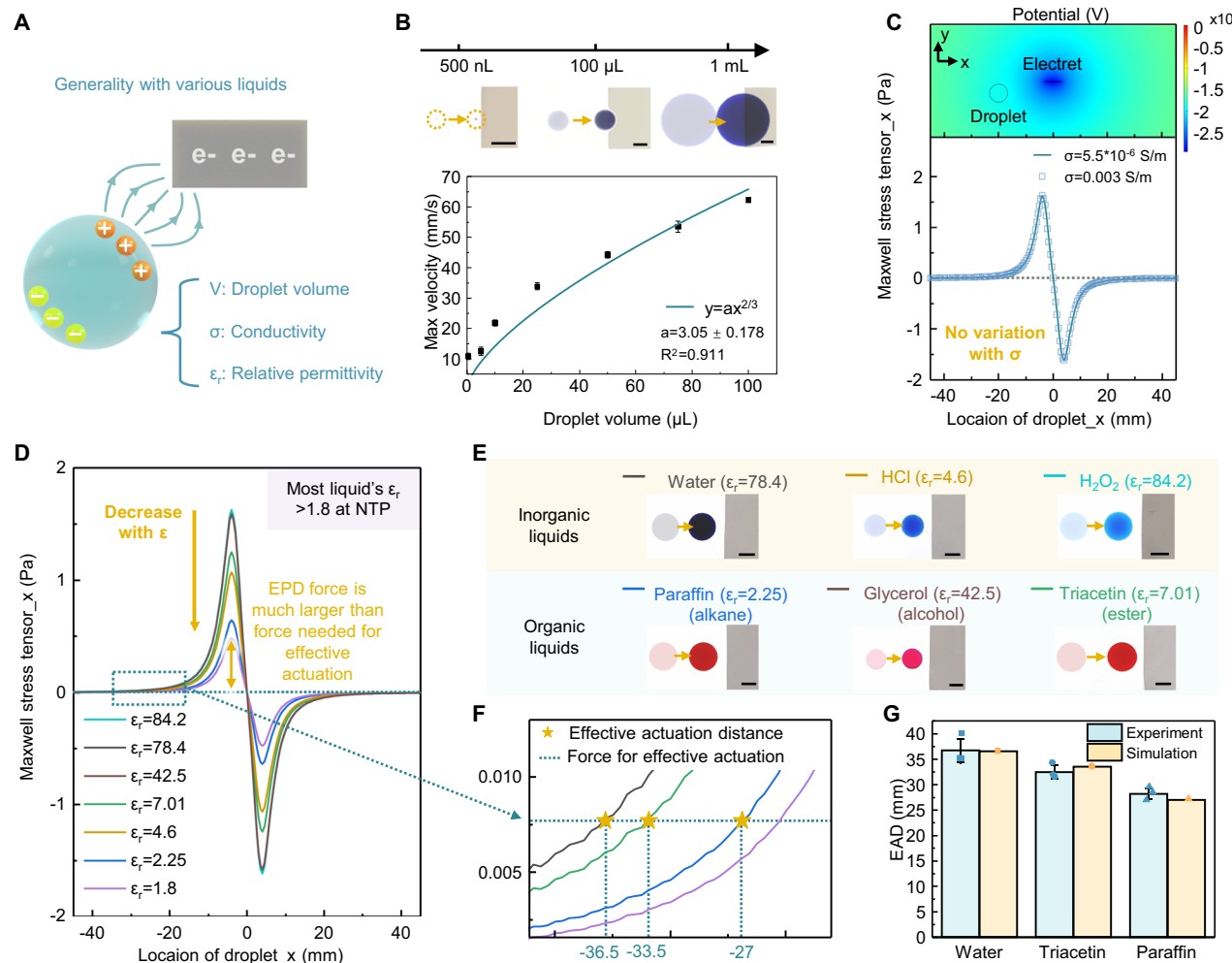

**Fig. 3 | Demonstration for EPD's generality with various liquids, including different droplet volumes, liquid conductivities, and liquid relative permittivities. A** Schematic diagram of EPD's generality with various liquids. **B** EPD-based actuation of droplets with volumes ranging from nanoliters to milliliters. The maximum actuation velocity increases with droplet volume under a two-thirds power relationship. Scale bars: 5 mm. Error bars, SD. **C** Simulated Maxwell stress tensor applied on droplets with different conductivities $\sigma$, showing that the generated EPD force applied on droplet does not change with conductivity. **D** Simulated Maxwell stress tensor applied on droplets with different relative permittivities $\varepsilon_r$ (solid lines). The calculated Maxwell stress tensor for the liquid with minimum relative permittivity of 1.8 at normal temperature and pressure (NTP) (purple line) is still much larger than the minimum force required for effective actuation (blue dash line), thus indicating that EPD has the potential to manipulate all types of liquids. **E** Optical images of EPD-based droplet actuation, including three common inorganic liquids as well as three common organic liquids, including alkane, alcohol, and ester. Scale bars: 5 mm. **F** Simulated effective actuation distance of water, triacetin, and paraffin (star symbol) obtained from the crossing between the calculated Maxwell stress tensor (solid line) and the minimum force required for effective actuation (dash line). **G** Comparison between the simulated and the measured effective actuation distance (EAD), demonstrating the validity of the simulation result. Error bars, SD ($n = 3$). Source data are provided as a Source Data file.

interactions with the surface adsorb undesirable proteins, hindering the droplet movement on EWOD[41–43]. Although the introduction of certain additives (e.g., Pluronics) may partially release the protein adsorption issue in EWOD, it still cannot fundamentally solve the problem since additives may be potentially cytotoxic and the species as well as the concentration of the surfactants need to be customized according to the operated protein solution[72]. On the contrary, when actuating droplets containing fluorescent proteins in EPD without any reagent additives, no significant change in fluorescence intensity is observed either on the trajectory or inside the droplet ($P > 0.05$) (Fig. 4C, Supplementary Movie 5). This result indicates that EPD-based droplet actuation will not cause observable protein residues, thus minimizing the risk of biofouling.

Besides body fluids and protein solutions, the effect of EPD on cell activity is also investigated (Fig. 4D). Compared to the control group in air, the percentage of living cells does not change significantly during the actuation process, demonstrating that EPD can maintain living cells' viability during actuation. In addition, the effect of EPD on the proliferation capacity of cells is also verified. Cells treated by EPD show a significant proliferation after 12-h incubation, where the concentration of living cells among them increases by 16.6% (Fig. 4E). In contrast, the number of living cells treated by the air-based EWOD decreased by 20.8% after incubating for 12 h. Similar negative effects on living cells are also observed in oil-based EWOD (Supplementary Fig. 11). These negative effects of EWOD on living cells may be caused by Joule heating and high electric field strength in the electrowetting effect[57,58], which are eliminated in the electrostatic charges-based EPD. Therefore, compared with either air-based or oil-based EWOD, EPD shows better compatibility with living cells. In addition to bio-samples, EPD-based droplet actuation is still feasible when the droplet contains other chemical samples, such as oil-in-water emulsions, water-in-oil emulsions, or nanoparticles-in-water (Supplementary Fig. 12).

The compatibility of EPD-based droplet actuation with various surroundings and substrates is also demonstrated (Fig. 4F). Droplet in

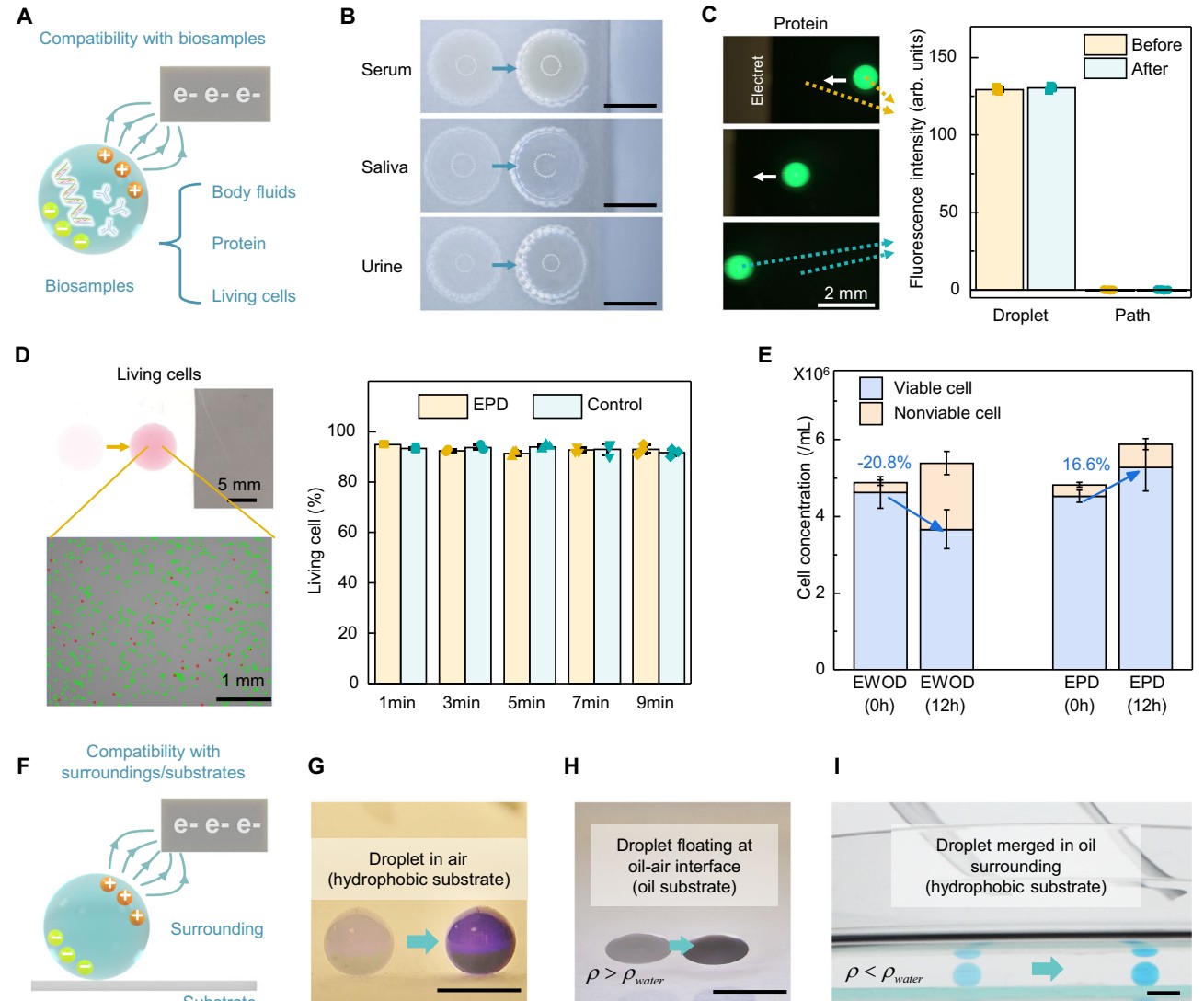

**Fig. 4 | Demonstration for EPD's compatibility with bio-samples and substrates. A** Schematic diagram of EPD's compatibility with bio-samples, including body fluids, protein, and living cells. **B** Optical images showing the EPD-based actuation of body fluids, including human serum, saliva, and urine. Scale bars: 3 mm. **C** Series of fluorescence microscope images showing the EPD-based actuation of fluorescent protein solution. The fluorescence intensity (in arbitrary units, arb. units) on the trajectory of the droplet or inside the droplet before and after droplet movement shows no significant difference (paired t-test, $P > 0.05$), demonstrating that no protein residue can be observed in the EPD-based droplet actuation. Error bars, SD ($n = 4$). **D** EPD-based actuation of living A549 cells within culture medium, where the percentage of living cells is analyzed and compared with the control group in air. Error bars, SD ($n = 3$). **E** Variation of viable and nonviable cell concentrations after being treated by EPD/EWOD for 30 min and then incubated for another 12 h, demonstrating the effect on cell proliferation capacity by EPD and EWOD. Error bars, SD ($n = 3$). **F** Schematic diagram of EPD's compatibility with various surroundings and substrates. **G** Optical images showing the EPD-based droplet actuation in air, on the hydrophobic surface. Scale bars: 3 mm. **H** Optical images showing the EPD-based actuation of droplet floating at oil-air interface, with an oil substrate which has a higher density $\rho$ than water. Scale bars: 3 mm. **I** Optical images showing the EPD-based actuation of droplet merged in oil surrounding which has a lower density $\rho$ than water, with a hydrophobic substrate. Scale bars: 3 mm. Source data are provided as a Source Data file.

the surroundings of air (Fig. 4G), oil-air interface (Fig. 4H), and oil (Fig. 4I) can all be actuated by EPD, as long as the permittivity of the droplet is different from that of the surrounding medium (Eq. (1)). As for the substrate, unlike EWOD[73,74], EPD has no requirements for substrate material or thickness, but it still requires a hydrophobic surface (Supplementary Fig. 13) or oil-substrate to reduce the actuation resistance and droplet residue. The actuation on hydrophilic surfaces would be limited by the increasing resistance (Supplementary Fig. 14, Supplementary Note 4). When oil substrate is utilized, the maximum actuation velocity can be further adjusted by introducing surfactant (Supplementary Fig. 15). The change of droplet shape from an approximate sphere to an ellipsoid may be the main reason for the increasing velocity[75,76]. Considering that droplet evaporation on the oil

substrate is only 30% of that on the hydrophobic substrate (Supplementary Fig. 16), the following quantitative experiments are all conducted by manipulating droplets floating on the oil-air interface instead of the hydrophobic substrate.

## Multiphysics droplet robotic system design
To fully automate the EPD-based droplet actuation, we further design a droplet robotic system, as shown in Fig. 5A and Fig. 1B. The proposed multiphysics system couples the EPD effect as well as electric and magnetic fields, consisting of three entities, including (1) a Printed Circuit Board (PCB) control matrix which can be programmed to generate localized magnetic field by powering specific coils; (2) EPD grippers made of magnetic responsive material and

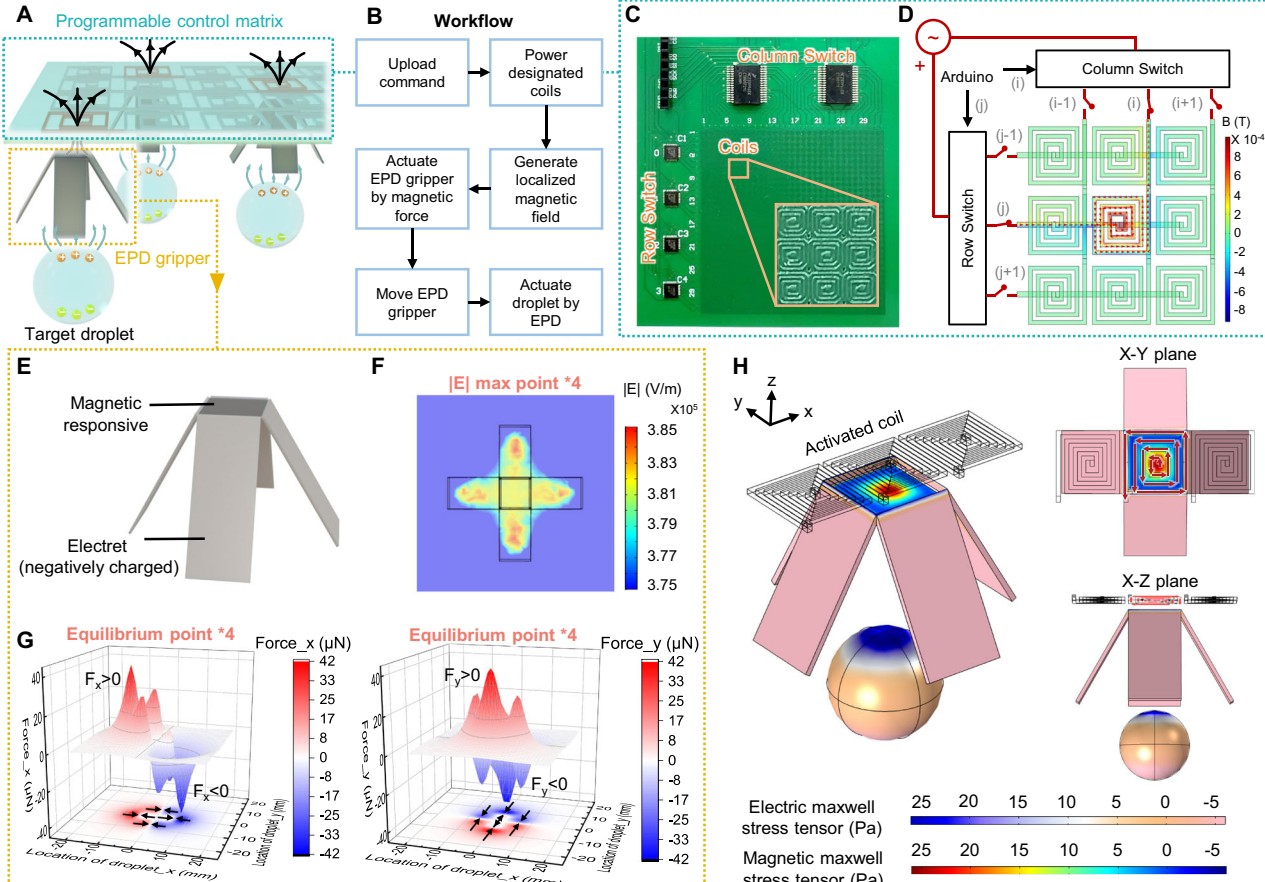

**Fig. 5 | Design of the multiphysics droplet robotic system. A** Schematic diagram of the proposed multiphysics droplet robotic system, consisting of three entities, including a programmable control matrix, EPD grippers, and target droplets. **B** The general workflow of the proposed multiphysics droplet robotic system. **C** Optical image of the programmable control matrix fabricated on a multilayer PCB, composed of row switches, column switches, electromagnetic coil matrix, and a signal/power socket. **D** Operation schematic of the control matrix and corresponding simulation result, which demonstrates the distribution of magnetic field when powering the coil at the coordinate of (j, i). **E** Schematic diagram of the EPD gripper, consisting of magnetic responsive material and electret composites. **F** Simulated electric field strength *E* of the EPD gripper (bottom view), in which four local maxima of the electric field are present. **G** Force generated by the EPD gripper in x direction (left) and y direction (right) on the droplet at different locations, with four force stable equilibrium points labeled with arrows. The direction of the force in x and y direction is consistent with the direction of the x and y coordinates, respectively. **H** Simulated magnetic Maxwell stress tensor on EPD gripper and electric Maxwell stress tensor on target droplet. Arrows represent the direction of current in coils. Source data are provided as a Source Data file.

electret composites; (3) target liquid droplet that can be polarized. To automate EPD-based droplet actuation, the command needs to be uploaded to the control board first (Fig. 5B). By powering the designated coil in the control matrix, a localized electromagnetic field can be generated. Then, the EPD gripper nearby will be actuated by magnetic force and move to the designated location. The shifted gripper subsequently attracts droplets via EPD so that the droplet can move along with it. By repeating this process, multiple EPD grippers can work collaboratively to actuate multiple droplets step-by-step to the target location.

In the programmable control matrix, electromagnetic coils are fabricated on a multilayer PCB and controlled by two integrated switches for row and column selection (Fig. 5C). By activating row and column switches corresponding to the specified coordinates, direct current will flow through the designated coils and generate a localized magnetic field (Fig. 5D). The strength of the generated vertical magnetic field is concentrated within the range of the specified coils. Therefore, it can accurately drive the magnetic responsive EPD grippers located near the specified coordinates without affecting those at longer distances, providing a foundation for multi-grippers synergistic cooperation. Compared with employing robotic arms, the approach of fabricating a control matrix on PCB reduces the fabrication cost of the

system (~US\$100 per set). It also offers the potential for spatially crossover manipulation so that multiple droplets can be processed in parallel.

As for the EPD gripper, magnetic responsive material and electret composites are utilized: The magnetic responsive section on the top can move the entire gripper to the designated coordinates under magnetic force; while the electret section at the bottom can generate a non-uniform electrostatic field to polarize and attract the target droplet below (Fig. 5E). Different from the slice-shape electret used in the above characterization set-ups, here the electret section is designed as a gripper shape to increase the number of local maxima of the generated electric field (Fig. 5F, Supplementary Fig. 17). Considering droplets will move towards the local electric field maxima under the EPD effect (Eq. (1)), such a gripper shape will enhance the stability of droplet actuation by providing multiple attraction points. The simulated EPD force applied on the droplet can further support this inference. In Fig. 5G, the arrow-labeled position where the direction of the force switches is the force stable equilibrium point of the droplet. Therefore, droplets nearby will tend to be attracted and stabilized at any of these points (Supplementary Fig. 18). Compared with the slice-shape electret, which only has one equilibrium point, the designed gripper shape can not only improve the stability of droplet actuation, but also

increase the force applied on the droplet and the effective actuation distance (Supplementary Figs. 19, 20, Supplementary Movie 6).

With the proposed programmable control matrix and EPD gripper, the operation of the entire system couples the multiphysics field. By converting the electricity into a programmable magnetic field, the control matrix exerts magnetic forces on the EPD grippers, while the charge carried by EPD grippers generates an electrostatic field to actuate the target droplet. The magnetic Maxwell stress tensor and electric Maxwell stress tensor acting in the entire system are shown in Fig. 5H. In the practical scenario, an extra actuation magnet can be used to amplify the electromagnetic field generated by the control matrix, balancing the weight of the EPD gripper, as shown in Fig. 1B.

Based on the design principles described above, by programming the control matrix, droplets can be actuated following different paths, such as the shapes of letters H, K, and U (Fig. 1C, Supplementary Movie 1). The actuation resolution is determined by the size of the coil, i.e., ~1.5 mm in this case, while the actuation precision is related to the movement of the actuation magnet, slightly lower than that of EWOD but comparable with a magnetic-based droplet actuation platform[12] (Supplementary Fig. 21). Other basic microfluidic functions like self-assembly, merging and mixing of multiple droplets can also be performed in the designed EPD-based droplet robotic system (Supplementary Fig. 22, Supplementary Movie 7). Although two droplets floating at the oil-air interface can gradually approach and eventually merge under capillary force (Supplementary Note 5)[77–81], the presence of an EPD gripper can further accelerate their approach and merging through attractive forces (Supplementary Fig. 23).

### Application of the EPD-based droplet robotic system for automating the scientific experiments

Leveraging the demonstrated EPD's generality for operable liquid types and compatibility with biochemical samples, the automated EPD-based droplet robotic system has the potential to impact the liquid-based scientific experiments in multiple fields. Here, we first apply the proposed droplet robotics system to automate bioassays for lithium detection in diverse biofluids as an example.

The detection of biomarkers/drugs in multiple biofluids can explore their metabolic relationships among various biofluids, thus contributing to non-invasive drug monitoring and precise medication[82–84]. However, electrical conductivity and protein concentration differ greatly among biofluids, such as human saliva, blood, and urine, and fluctuate over a wide range due to individual differences (Fig. 6A)[85–89]. For conventional liquid actuation techniques such as EWOD, the difference in liquid conductivities among different biofluids will affect the setting of working voltage and actuation robustness[70,71]. The proteins within will also adsorb on the surface due to electrostatic interaction and hydrophobic interaction, resulting in the sample immobilization or cross-contamination[41,42]. In experiments, most biofluids cannot move smoothly on the tested EWOD platform, while the performance improvement brought by increasing operating voltage (Supplementary Fig. 24) or introducing oil surrounding (Supplementary Figs. 3 and 8) also varies significantly among different biofluids. Therefore, in practical applications, actuation of various biofluids on EWOD needs to either dilute the biofluids[90], remove the protein in it[91], or introduce a certain amount of surfactant[92], while also modifying the voltage and frequency settings[93–95]. On the contrary, according to our characterizations (Figs. 3, 4), liquid conductivity and protein concentration will not have a significant effect on the EPD-based droplet actuation. When three biofluids (serum, saliva, urine) are tested on the proposed EPD-based droplet robotic system, all the tested samples can move smoothly (Fig. 6B). Even without altering any parameter settings, their actuation performances show no obvious difference.

To assist the process of bioassay, a microfluidic detection chip is designed and powered by the EPD-based droplet robotic system (Fig. 6C). The detection chip is designed to perform three detections in

parallel, and here we take two calibrations and one sample detection as an example to demonstrate. On the detection chip, three reagents loading areas are divided to load masking, probe, and buffer solution, respectively. Three working regions are also designated for merging, mixing, and reacting these reagents with calibration samples/testing sample, respectively. In the initial state, two calibration samples of known lithium concentrations and the masking solution are preloaded (Fig. 6D). After loading the tested bio-sample, three EPD grippers are programmed to work collaboratively to implement the steps of the automated assay, including sample preparation, calibration 1, calibration 2, and sample detection. The tasks executed by each EPD gripper and their trajectories are listed step-by-step, along with representative screenshots, as shown in Fig. 6E and Supplementary Movie 8.

Particularly, EPD grippers 1 and 2 are in charge of transporting and mixing the tested bio-sample with the masking solution, shielding other interfering ions in the bio-sample. EPD grippers 2 and 3 are responsible for capturing the injected sub-droplets of buffer and probe solutions and mixing them with the prepared calibration sample/tested bio-sample. Lithium ions within the sample will bind to the probe after dilution, thereby shifting the absorbance profiles quantitatively. In this way, two in-situ calibrations and one bio-sample detection can be performed automatically, and lithium concentration in the tested bio-sample can be calculated based on the established calibration curve (Supplementary Fig. 25). The tested human serum, saliva, and urine lithium concentrations show no statistically significant difference from the reference values ($P > 0.05$, Fig. 6F), demonstrating the reliability of the bioassay results accomplished by the EPD-based droplet robotic system. In addition to measuring the absorbance at specific wavelengths, similar standard curves and detected results can also be derived by simple in-situ photography and RGB analysis, offering the feasibility of further improving the integration of the system (Supplementary Fig. 26).

Besides the application example of automating a bioassay for diverse biofluids, EPD is also validated for establishing cell-bacteria models with dynamic monitoring. As a demonstration, we establish an in vitro cell-bacteria model of inflammation and in situ detect the generated inflammatory mediator, IL-1β, on the EPD-based system (Supplementary Fig. 27). By dynamically repeating bacterial infection of cells with the EPD-based droplet robotic system, our result validates the non-monotonic relationship between the concentration of inflammatory mediators and repeated bacterial infections, which can help to explore the generation of inflammatory mediators and the connection between diseases and inflammatory mediators[96–98].

In the two applications demonstrated above, droplets of volumes ranging from 5 μL to 1 mL are manipulated, while six different solutions, living cells, living bacteria, and three kinds of body fluids are also operated, taking full advantages of EPD's superior liquid operability and bio-compatibility. Furthermore, the proposed system also exhibits the ability of spatially crossover manipulation, thus endowing the potential to work in parallel with multiple robots. Compared with other workstations that repeat the same experimental steps in a linear temporal manner, our EPD-based droplet robotics system thus demonstrates a higher flexibility. Therefore, based on the demonstrated applications, the proposed EPD-based droplet robotic system shows wide applicability and impact on scientific research, with the potential for further applications in multiple fields that require precise liquid manipulations.

## Discussion

In this study, we introduce a droplet robotic system based on the mechanism of EPD to address the compatibility issues of the existing droplet actuation platforms with liquid types and biochemical samples. The proposed EPD mechanism utilizes electret material to generate a non-uniform electrostatic field, polarizing and attracting various liquid droplets. Compared with the traditional liquid

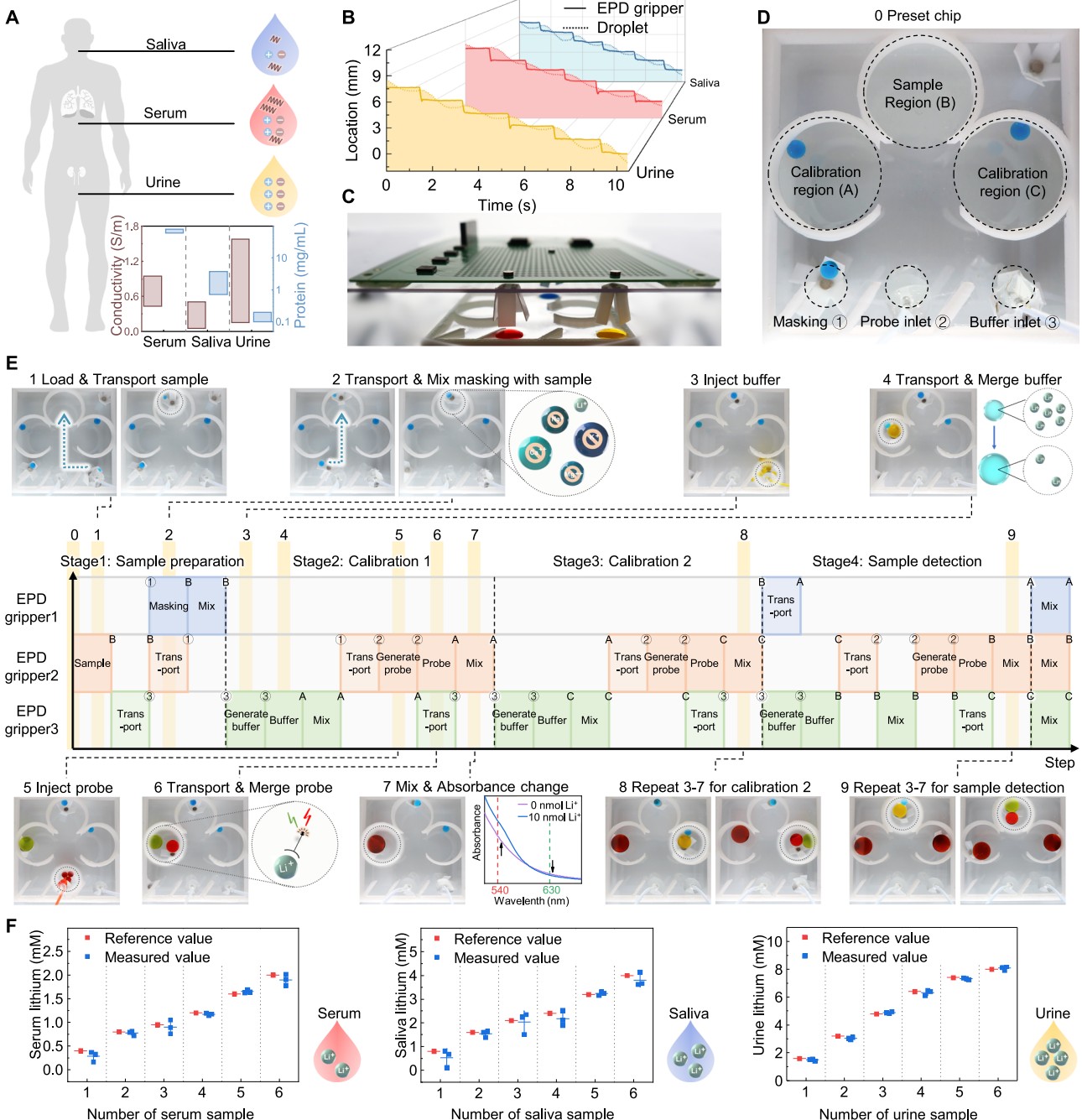

**Fig. 6 | Automated lithium detection in serum, saliva, and urine conducted by the EPD-based droplet robotic system. A** Schematic diagram of different body fluids (serum, saliva, and urine), demonstrating that their conductivity and protein concentration vary greatly[85–89]. **B** Characterization of serum, saliva, and urine's actuation on the EPD-based droplet robotic system. The solid line shows the trajectory of the EPD gripper while the dash line shows the trajectory of the actuated biofluids. **C** Optical image of the EPD-based droplet robotic system and the auxiliary detection chip. **D** Optical image of the microchip setup, in which three detections can be performed in parallel. Dyed droplets are used here instead of transparent ones for visualization purpose. **E** The step-by-step workflow of the EPD-based droplet robotic system when performing lithium detection, demonstrating the tasks executed by each EPD gripper and their trajectories along with representative screenshots. **F** Comparison between the tested lithium results of serum, saliva, and urine samples and the reference values, in which the testing values show no statistically significant difference to reference values (One sample t-test, $P > 0.05$), demonstrating reliability of the bioassay results accomplished on-chip. Error bars, SD ($n = 3$). Source data are provided as a Source Data file.

polarization generated by AC/DC electric field, EPD employs the intrinsic electrostatic charges carried by the electret instead. Therefore, EPD does not generate conduction current and differs at the level of equivalent circuit models (Supplementary Note 1), leading to different variables and application scenarios. Thus, the EPD mechanism complement the existing principle of liquid polarization from an electrostatic perspective.

Benefiting from the novel EPD mechanism, we validate the EPD-based droplet actuation with superior adaptability with liquid types and biochemical samples, while also achieving full automation based on a multiphysics control system. Compared with the existing droplet actuation platform, the proposed EPD-based droplet robotic system exhibits a high generality of operable liquid types (various inorganic/organic liquids with relative permittivity ranging from 2.25-84.2 and

volume ranging from 500 nL-1 mL), high compatibility with biochemical samples (multiple body fluids, proteins, and living cells), high compatibility with substrates and surroundings (air/oil/air-oil interface), high actuation speed (up to 60 mm/s), low working voltage (5.5 V), and low cost of fabrication (lower than US$1 for consumables and ~US$100 for control system) (Supplementary Table 1). We further apply the EPD-based droplet robotic system to automate lithium detection for diverse biofluids and establish in vitro cell-bacteria models with dynamic monitoring. In the demonstrated applications, six different solutions, living cells, living bacteria, and three kinds of body fluids are manipulated, fully certifying the impact and applicability of EPD in multi-disciplinary scientific research that require precise liquid manipulations.

In future, to further enhance its adaptability in multi-disciplinary applications, the EPD-based droplet robotic system can be potentially developed from both technical and application perspectives. From the technical perspective, the system could be miniaturized and modularized to achieve fluid manipulation at the nanoscale. The system we show in the paper is mainly designed to manipulate droplets on a μL scale (500 nL−1 mL). If we downsize the system while maintaining the original design, the charge density of the electret needs to be increased to ensure functionality (Supplementary Fig. 28). Besides, we can also microfabricate a 2D charge distribution-controllable electret material. This approach has the potential to actuate droplets by directly programming the local electric field distribution without physically moving the electret material, thus further reducing the size of the system. Another promising area of technical development lies in the improvement of controllability and speed. At the actuation system level, the magnetic force generated by the programmable control matrix can be further enhanced by increasing the current (e.g., replacing switch ICs with higher current threshold chips) or modifying the coil design (e.g., more layers of coils), increasing system robustness and switching frequency. At the actuation material level, we can utilize different electret materials (e.g., CYTOP, which can provide a higher surface charge density up to 2 mC/m² for a 15-μm thick film[99]) or apply different charging methods (e.g., corona charging or electron-beam irradiation[100]) to further increase the surface charge density of the electret, providing a larger actuation force (Supplementary Note 2). In terms of application advancements, the proposed droplet robotic system provides new access to accomplish high-throughput and high-precision experiments, promoting experimental efficiency in future. The potential incorporation of machine learning and artificial intelligence (AI) into our system can help analyze massive amounts of data obtained from automated systems, unearthing the hidden correlations among various parameters and results. Besides, the automated system also offers the possibility of remote experiments, especially for experiments involving hazardous operations and harmful chemicals. This approach of remote experimentation also lays the foundation for promoting communication and cooperation between laboratories, facilitating the real-time sharing of research results.

## Methods

### Materials
HFE-7500 (Novec Engineered Fluid, 3 M) was used as the oil substrate for EPD-based droplet actuation. 0.05% surfactant (Pico-Surf, Sphere Fluidics) could also be added in HFE-7500 to adjust the shape and surface area of the droplet floating on HFE surface. As for the hydrophobic substrate, superhydrophobic coating (204A, Wateroff) was spin-coated onto the polyethylene terephthalate (PET) film sheets, resulting in a contact angle of 163° after drying for 24 h (Supplementary Fig. 13). Oil substrate was used for most of the experiments unless otherwise specified. DI water, HCl (Aladdin), H₂O₂ (QuantaRed Stable Peroxide Solution, Thermofisher), Paraffin (Aladdin), Glycerol (Sigma-Aldrich), Triacetin (Aladdin), Hexadecane (Sigma-Aldrich) were actuated by EPD respectively. Methylene blue (TCI), oil red O (C.1.26125, Aladdin), and

rhodamine B (Aladdin) were added to visualize the movement of transparent droplets in some optical image demonstrations.

### Preparation of electret and charge measurement
Negatively charged electret made of PTFE was charged by contact electrification with copper, while positively charged electret made of glass was charged by contact electrification with PTFE. By varying the applied friction, the amount of charge obtained by the electret can be adjusted accordingly. To measure the charge amount possessed by electret, electret was placed within the Faraday cup connected with the programmable electrometer (6514, Keithley Instruments model) under the charge measurement model.

### Characterization of the EPD-based droplet actuation
In characterization experiments, one slice of electret (35 × 30 × 0.2 mm) was used to actuate the droplet of 20 μL floating on the oil surface, unless otherwise specified. The experiments were independently replicated for at least three times. To quantitatively characterize the process of EPD-based droplet actuation, the movement of droplet was analyzed with the software Tracker to calculate its location, velocity, and acceleration. Dyes were introduced to assist the tracking of target droplet. Specifically, to measure the effective actuation distance, the charge amount possessed by the electret was measured at the beginning of each test, and then attached to the self-constructed slider to approach the droplet horizontally. The height difference between electret and droplet is controlled as 10 mm. When the droplet started to move after the electret moved to a certain distance, the horizontal distance between the two at this point was defined as the effective actuation distance. To measure the minimum and maximum height differences for required effective actuation, surface charge density of the electret was also measured first, and then attached to the self-constructed slider to approach the droplet vertically. The electret was repeatedly moved laterally within about 10 mm from the droplet as it approached the droplet vertically. When the droplet began to move with the electret at a certain height, this height was defined as the maximum height difference required for effective actuation. As the electret continued to descend to a certain height and the droplet was absorbed to the electret and contact occurred, this height was then defined as the minimum height difference required for effective actuation. To measure the max velocity of droplet actuation, electret after charge measurement was attached to the self-constructed slider at 5, 8, 11 mm above the droplet. The electret accelerated laterally and actuated the droplet along with it. The maximum velocity of the droplet was analyzed by Tracker.

### Simulation of the EPD-based droplet actuation
Simulation of the EPD-based droplet actuation was conducted using finite element analysis tools (COMSOL Multiphysics 5.4). Several simplifications were made during modeling and the scenario was simplified to 2D. The surrounding environment of air was modeled as a square shape with a dimension greatly larger than the electret and droplet, and all the surface boundaries were set grounded. The electrostatic interactions in the system were considered dominating, thus other physical phenomena were ignored in the simulation. According to the measured experiments parameters, the electret was set as a rectangle of 6 × 2 mm, with surface charge density of $-1.74 \times 10^{-5}$ C/m², while the droplet was set as a circle with radius of 3 mm in most cases, unless otherwise specified. To analyze the average Maxwell stress tensor applied on droplet, line averages of the x component of the Maxwell stress tensor applied on the circular edge of the droplet were calculated. In Fig. 2E, to quantify the effect of droplet's charge density, the droplet was set to carry a spatial charge density of 0 and $\pm 2 \times 10^{-4}$ C/m³ according to the measurement[59], respectively. The average Maxwell stress tensor applied on droplet was calculated with various droplet's location. The electret was set to locate at 0 mm.

## Human body fluids sample preparation

All human urine and saliva samples were obtained following The University of Hong Kong, Human Research Ethics Committee approved research project, HREC No. EA230092 with consent from all participants. Upon collection, urine and saliva was centrifuged at $4830 \times g$ for 20 min and the supernatant was frozen at −20 °C. The human serum (H3667) was purchased from Sigma-Aldrich, which is heat inactivated, from human male AB plasma, USA origin, sterile-filtered. The human serum was frozen at −20 °C in small aliquots until used.

## Protein actuation experiments

Droplet of fluorescent protein (FITC-labeled goat anti-rabbit IgG (H + L), Beyotime) floating on oil surface was actuated by EPD, while the whole process was observed and recorded by fluorescence microscopy. To analyze the fluorescence intensity of the droplet and its moving path, mean gray value of four different points are calculated via ImageJ in droplet, moving path, and background, respectively. The average value in background is then subtracted from the value of droplet and moving path.

## Living cells actuation experiments

A549 cells, obtained from the American Type Culture Collection (CCL-185), were utilized in living cells actuation experiments. First, A549 cells were maintained in DMEM supplemented (10569010, Gibco) with 10% fetal bovine serum (10099141, Gibco) and 1% penicillin–streptomycin (15070063, Gibco). Cells were then incubated at 37 °C, 5% $CO_2$ for 48 h and harvested for actuation experiments. To demonstrate the cell viability during actuation process, three droplets containing A549 cells and the culture medium were actuated by EPD. 10 μL from each cell suspension droplet was extracted and mixed with 10 μL of 0.4% Trypan blue stain (15250061, Gibco) homogeneously after being actuated for 1, 3, 5, 7, 9 min and counted by Countess II FL Automated Cell Counters (Invitrogen, Thermo Fisher Scientific corporation), respectively. Three control droplets were also dropped on the petri dish and exposed to air together with the experimental group. 10 μL from each droplet was also extracted and mixed with 10 μL of Trypan blue stain homogeneously at 1, 3, 5, 7, 9 min and counted accordingly. The effect of EPD on cell proliferation was also tested. After treating 300 μL droplets containing A549 cells with EPD for 30 min, a pipette was used to flush the droplets repeatedly to distribute the cells evenly within the droplet. Droplets containing cells were then extracted and incubated for another 12 h. 10 μL from each droplet was extracted and mixed with 10 μL of Trypan blue stain homogeneously before and after the whole experiments, repeated for three times. Viable and nonviable cell concentrations for each test were then calculated by Countess II FL Automated Cell Counters.

THP-1 cells, obtained from the American Type Culture Collection (TIB-202), were utilized in experiments of analysing the impact of EPD system setup on cell culture. First, THP-1 cells were maintained in RPMI 1640 Medium (11875093, Gibco) with 10% fetal bovine serum (10099141, Gibco) and 1% penicillin–streptomycin (15070063, Gibco). Cells were then incubated at 37 °C, 5% $CO_2$ for 48 h and harvested for experiments. Then, three 300 μL droplets containing THP-1 cells and the culture medium were placed on EPD system and cultured for 24 h without power supply. After 24 h, 10 μL from each cell suspension droplet was extracted and mixed with 10 μL of 0.4% Trypan blue stain homogeneously, repeated for three times. Viable and nonviable cells concentration for each test were then calculated by Countess II FL Automated Cell Counters.

## EWOD comparison experiments

Two pieces of ITO glass were used to form the upper and lower plates of EWOD. The upper plate was then coated with a 50 nm Teflon-AF layer by spin-coating the solution (1 wt% in FC-40, 3 M, USA) at 1500 rpm for 60 s (AC-200SE, Lebo Science). The ITO electrode pattern on lower plate was fabricated by photolithography using negative photoresist (SU-8 2025, MicroChemicals) on mask aligner (MA/BA6, SUSS)[71]. The lower plate was deposited with a 3.5 μm thick parylene-C film using low-pressure chemical vapor deposition (CVD) equipment (LH300, LaChi Enterprise), then a Teflon-AF solution (1 wt% in FC-40, 3 M, USA) was spin-coated on the parylene-C film at 1500 rpm for 60 s. The assembled plate was baked at 165 °C on a hotplate for 15 min. The amplitude of the voltage used when applying the ITO glass on the digital microfluidic platform is from 60 V to 150 V, with the frequency of 1000 Hz.

The fabricated EWOD device was demonstrated to compare the droplet actuation performance with EPD. Water, glycerol, triacetin, human serum, human saliva, human urine was tested on the fabricated EWOD, respectively. Specifically, 1.8 μL droplet was used in the air-based EWOD, and 1 μL droplet with 0.5 μL Decamethyltetrasiloxane (Aladdin) was used in the oil-based EWOD. To quantitatively analyze the actuation of these liquids, the software Tracker was used to track the front edge of the droplet and calculate its real-time position based on the recorded videos. The actuated distance of the front edge of the droplet normalized by the distance between two neighboring electrodes (for EWOD) or coils (for EPD) was defined as the relative moving distance (Supplementary Figs. 2, 3). 0.05 mg/mL FITC-BSA solution was also tested on EWOD, where the protein adsorption on EWOD is observed from fluorescence microscopy.

Living cells in culture medium was also tested on EWOD. Due to protein adsorption, the droplet cannot be actuated directly on the double-plate EWOD. Therefore, we simplified the EWOD device utilized for comparing droplet actuation performance as a pair of ITO glass plate electrodes (50 × 50 × 1.0 mm) to investigate how the electric field applied between EWOD's upper and lower plate will affect living cells. Two ITO glass plate electrodes sandwiched the insulated rubber tape with a thickness of 1 mm as a spacer. The coating process for the ITO glass was the same as that mentioned above. To analyze the impact of electrowetting effect on cell proliferation, 300 μL droplets contained living A549 cells were added between the electrodes for air-based EWOD, while 50 μL Decamethyltetrasiloxane was additional introduced for each droplet on oil-based EWOD. DC voltage of 150 V was applied between the two electrodes for 30 min. After that, a pipette was used to flush the droplets repeatedly to wash down the cells attached to the substrate and to distribute the cells evenly within the droplet. Droplets containing cells were then extracted and incubated for another 12 h. 10 μL from each droplet was extracted and mixed with 10 μL of Trypan blue stain homogeneously before and after the whole experiments, repeated for three times. Viable and nonviable cells concentration for each test were then calculated by Countess II FL Automated Cell Counters. To analyze the impact of oil-based EWOD system setup with cell culture, 300 μL droplets contained living THP-1 cells and 100 μL Decamethyltetrasiloxane were added between the electrodes for oil-based EWOD and cultured for 24 h without power supply. 10 μL from each droplet was extracted and mixed with 10 μL of Trypan blue stain homogeneously before and after the whole experiments, repeated for three times. Viable and nonviable cells concentration for each test were then calculated by Countess II FL Automated Cell Counters.

## EPD-based multiphysics droplet robotic system

The programmable control matrix was established on a muti-layer PCB refer to the actuation concept of "Ferrobotic system"[12], which contained 32 × 32 coils matrix, four row switches MAX14662 (Maxim Integrated), two column switches MC33996 (NXP Semiconductors), and a 1 × 10 pin header for power supply and communications with an Arduino Uno. The coils were consisted of three turns of 1 mm wide wire and stacked three layers on PCB, powered by a 0.2 A current. By uploading codes through computer to the Arduino Uno, it could

selectively turn on the specific row switch and column switch, powering the coil at the designated coordinate and generate a local electromagnetic field.

The EPD gripper was made of a hybrid of electret (PTFE) and magnetically responsive material (Neodymium-Iron-Boron magnet, DH101, 1/32 inch in thickness and 1/10 inch in diameter, K&J Magnetics). The electret material formed the shape of the gripper through origami, while the magnetically responsive material is adhered to the bottom of the gripper, providing the ability to be driven by magnetic field. In the EPD-based droplet robotic system, the EPD grippers were suspended beneath the control matrix by magnetic force. To balance the self-weight of the gripper, an actuation magnet is also utilized and placed above the control matrix. It could enhance the localized magnetic field generated by coils and provide extra attracting force to the EPD gripper. When switching on coils at different coordinates, the actuation magnet and the suspended EPD gripper could be actuated simultaneously through magnetic force. To minimize friction as the EPD gripper moves, a smooth pad (PET and glass sheet) was placed between the gripper and the control matrix.

### Simulation of the multiphysics droplet robotic system

Simulation of the multiphysics droplet robotic system was conducted using finite element analysis tools (COMSOL Multiphysics 5.4). Several simplifications were made during modeling. The surrounding environment of air was modeled as a cube shape with a dimension greatly larger than the coils and grippers, and all the surface boundaries were set grounded. To simulate the magnetic field generated by control matrix, the electromagnetic interactions in the system were considered dominating, thus other physical phenomena were ignored in the simulation. The three-layer coils were also simplified as a one-layer coil, and the inlets of the coils in each row are connected, while the outlets of the coils in each column are also connected (Fig. 5D). Results of the simulation were used to qualitatively characterize the distribution of the magnetic field. To simulate the electrostatic interaction between the EPD gripper and the droplet, electrostatic field physics was used. The simulation used the same EPD gripper dimensions as the experimental setup, and the surface charge density of the electret material was set to be $-1.6 \times 10^{-5}$ C/m$^2$. To calculate the force applied on droplet, the x and y component of the Maxwell stress tensor were integrated on the surface of the droplet (Fig. 5G). In the simulation of the whole system, magnetic and electrostatic field physics were utilized simultaneously. The actuation magnet was ignored during simulation, and the magnetic Maxwell stress tensor applied on the EPD gripper through the activated coil as well as the electric Maxwell stress tensor applied on the droplet through EPD gripper is demonstrated, qualitatively profiling the overall multiphysics coupling in the whole system.

### Microfluidic detection chip

The microfluidic detection chip was 3D printed from photosensitive resin, with 7 mm in height and 50 mm in length and width. A super-hydrophobic layer (NeverWet) was coated on the wall of the chip (Supplementary Fig. 29). In use, the chip was glued to the petri dish with Epoxy (Devcon) and filled with HFE to provide oil substrate. The demonstrated chip was designed to perform three tests in parallel, including three working regions and three reagent loading areas (Fig. 6D). For probe and buffer loading area, considering these reagents need to be loaded repeatedly, two microtubes with a diameter of 0.034″ I.D. × 0.052″ O.D. (LDPE, Scientific Commodities) were inserted at the bottom of the chip, and the inlet of the microtube located beneath the HFE-air interface. The microtubes were connected to two syringes installed on pumps (SPLab01, DK Infusetek). The pre-loaded probe solution and buffer solution were infused to the chip, and the generated sub-droplets of buffer/probe would rise to the surface of the HFE due to buoyancy and collected by the EPD grippers above.

### Implementation of the lithium detection in multiple bio-fluids with EPD-based droplet robotic system

To conduct calibration tests, 5 μL of the prepared bio-sample with known lithium concentration (LiCl solution spiked human serum with concentrations of 0 μM, 400 μM, 800 μM, 1200 μM, 1600 μM, 2000 μM; LiCl solution spiked human saliva with concentrations of 0 μM, 800 μM, 1600 μM, 2400 μM, 3200 μM, 4000 μM; LiCl solution spiked human urine with concentrations of 0 μM, 1600 μM, 3200 μM, 4800 μM, 6400 μM, 8000 μM) was mixed with 15 μL of masking solution (ab235613 lithium assay kit, Abcam) to form the calibration sample. These concentrations were selected based on the clinically empirical range of lithium in various human body fluids[101]. Due to the higher lithium concentration in human saliva and urine, the prepared saliva and urine lithium samples were diluted two-fold and four-fold, respectively, before mixing with masking solutions. In each calibration process, the prepared calibration sample needed to be mixed with 130 μL buffer and 100 μL probe solution (ab235613 lithium assay kit, Abcam). For each real bio-sample detection process, the real sample of human serum/saliva/urine were spiked with LiCl solution (ab235613 lithium assay kit, Abcam) with the concentration serving as a reference value. The prepared bio-sample needed to be mixed with 15 μL of masking solution (ab235613 lithium assay kit, Abcam) first, and then mixed with 130 μL Buffer and 100 μL probe solution during sample detection process. Among them, buffer and probe solution were loaded by pump and microtubes connected to the buffer and probe loading area each time, while the masking, calibration samples, and tested bio-sample were directly preloaded inside of the chip before test. The capture of the generated sub-droplet, transportation of reagents to the designated working area, merging and mixing of the reagents with samples were conducted by the EPD-based system. Each detection was repeated at least three times with independent samples, and the practical process was not necessarily performed according to the combination of two calibration and one test sample.

To obtain the tested result, the mixed droplet can be either extracted and transferred to a 96-well plate to measure absorbance at 540 nm and 630 nm by microplate reader (Spectramax iD5, Molecular Devices) or simply analyzed through in-situ photography and RGB analysis, as shown in Supplementary Fig. 26. To obtain the lithium standard curve in Supplementary Fig. 25, we subtracted the absorbance ratio ($OD_{540}/OD_{630}$) of the reagent blank (0 μM) from the measured absorbance ratios of other calibration samples. Then we plotted the background-subtracted absorbance ratio values of all calibration samples and calculated the slope of the standard curve. This standard curve could be used to calculate the measured value of lithium concentration for the tested bio-sample.

### Step-by-step demonstration of the workflow of the EPD-based droplet robotic system for automated lithium detection

To demonstrate the step-by-step workflow, a group of tests consists of two calibrations and one real bio-sample detection is performed as an example. Dyed droplets are used here instead of transparent real samples/reagents for clear demonstration. The operation process of the system was filmed with a camera (PowerShot G7X Mark III, Canon) from bottom up with an upward view (Fig. 6E and Supplementary Movie 8). The video was filmed and edited in segmental settings to demonstrate the detailed step-by-step workflow and offer clear operation details. In the demonstration, droplets representing masking, calibration samples, and tested bio-sample were directly preloaded inside of the chip, while the solution representing buffer and probe were loaded by pump and microtubes. Three EPD grippers were programmed to work collaboratively to capture the generated sub-droplet, transport reagents to the designated working area, merge and mix of the reagents with samples automatedly. Specifically, according to Fig. 6E, in the sample preparation stage (steps 0–2), after loading the tested bio-sample to the chip, EPD grippers 1 and 2 would transport

the sample and masking solution to sample region for mixing. This step was to shield other ions in the bio-samples from influencing the test results. In the second stage of Calibration 1 (steps 3–7), buffer and probe solutions were injected into the chip by pump, respectively (steps 3 and 5). Meanwhile, EPD gripper 2 and 3 would capture the generated sub-droplets and transport them to the calibration region (A) to mix with the prepared calibration sample (steps 4 and 6). Lithium ions within the sample would bind to the probe after dilution, thereby quantitatively shifting the absorbance profiles (step 7). Similarly, the generation, transporting, and mixing process of buffer and probe solutions were repeated for the third stage of Calibration 2 and fourth stage of Sample Detection, respectively (steps 8 and 9).

### Establishment of in vitro cell-bacteria model of inflammation and in-situ detection of inflammatory mediator with EPD-based droplet robotic system

To prepare the original bacteria and cells for experiments, *Escherichia coli* (*E. coli*; ATCC 25922) were cultured with Luria-Bertani (LB) medium at 37 °C overnight. Then, *E. coli* was harvested and washed three times with PBS using centrifugation ($1207 \times g$, 5 min). The obtained bacterial cells were resuspended and diluted to about $6 \times 10^7$ CFU/mL in PBS for experiment. THP-1 cells were maintained in RPMI 1640 Medium (11875093, Gibco) with 10% fetal bovine serum (10099141, Gibco) and 1% penicillin–streptomycin (15070063, Gibco). THP-1 cells were then incubated at 37 °C, 5% $CO_2$ for 48 h. After that, the contents were transferred to a centrifuge tube and spun at $300 \times g$ for 4 min. The cell pellet was resuspended in fresh RPMI 1640 Medium for experiment.

To establish the in vitro cell-bacteria model of inflammation with dynamic monitoring on the EPD system, 995 μl droplets containing $3 \times 10^6$ /mL THP-1 cells and 5 μL droplets containing $6 \times 10^7$ CFU/mL *E. coli* were added. Three groups of cell droplets and bacteria droplets were merged and mixed by the EPD system individually, and subsequently incubated for 12 h. Then, one of the mixed cell-bacteria droplets was observed under a microscope, labeled as the 1st round infection sample. The other two mixed cell-bacteria droplet were introduced with a new bacteria droplet, respectively, and then incubated for another 12 h. After repeating same processes for the second and third droplet, we acquired the 2nd round infection sample and the 3rd round infection sample, respectively. The concentration of generated human IL-1β in each sample could be detected by introducing 333 μl of the mixed antibody solution (containing 1.2 nM Eu-labeled anti-hIL1β Antibody and 12 nM ULight labeled anti-hIL1β Antibody, TRF1220C, LANCE Ultra IL1β (Human) Detection Kit, Revvity) to each sample immediately after sample acquisition. The mixed droplets were then incubated for 60 min on the EPD system.

After processing the samples and bioassay steps on the EPD system, the final droplets were extracted and centrifuged under $1207 \times g$ for 5 min to read the measurement value. The obtained supernatants were then transferred to a 96-well plate (60 μL per well) to conduct TR-FRET measurement with an excitation wavelength of 350 nm and an emission wavelength of 665 nm (Spectramax iD5, Molecular Devices). To obtain the human IL-1β standard curve, standard dilutions were prepared with fresh RPMI 1640 Medium and reconstituted hIL1β (TRF1220C, LANCE Ultra IL1β (Human) Detection Kit, Revvity). After mixing 45 μL of the standard dilution with 15 μl of the prepared antibody solution and incubating them for 60 min, same TR-FRET measurements were conducted, and the results were fitted by a sigmoidal dose-response curve.

### Evaluation rules of the comparisons between EPD-based droplet robotic system and other existing automated droplet actuation system

The comparison in Fig. 1D was conducted based on the detailed information in Supplementary Table 1, which was either based on the results of our experiments (labeled with the number of the figure in which relative data are presented) or based on the references (labeled with the number of the specific reference). For the perspectives of 1/working voltage and 1/cost of fabrication, the lower working voltage and cost of fabrication in the Supplementary Table 1, the higher the evaluation result in Fig. 1D. For the perspectives of generality with operable liquids, compatibility with bio-samples, and compatibility with substrates & surroundings which included multiple sub-indexes in Supplementary Table 1, their evaluation results were calculated by averaging the evaluations of corresponding sub-indexes.

### Reporting summary

Further information on research design is available in the Nature Portfolio Reporting Summary linked to this article.

## Data availability

Data generated in this study are provided in the paper and/or the Supplementary Information. Source data are provided with this paper.

## Code availability

The code for the general control of the programmable control matrix is available at: https://github.com/zrthelenhku/Dropletrobots/tree/main.

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

## Acknowledgements

We are grateful to use SUSTech Core Facilities for the fabrication of the EWOD upper and lower plate and providing the access to mask aligner. We thank Dr. Xing Cheng at the Department of Materials Science and Engineering, Southern University of Science and Technology for providing the access to CVD equipment, digital microfluidics platform, and fluorescence microscope. We thank Mr. Miao Xu for providing the living *E. coli*. We'd also like to thank Dr. Yang Cao, Dr. Xiaolai Li, Dr. Wei Li, and Mr. Tianshu Jiang for the discussion. This work was supported

by the General Research Fund (Nos.17307919, 17303123) from the Research Grants Council of Hong Kong, and the Excellent Young Scientists Fund (Hong Kong and Macau) (21922816) from the National Natural Science Foundation of China (NSFC), which were all received by H.C.S. H.L. acknowledges support from the NSFC Young Scientists Fund (Hong Kong) (No. 32201181), General Research Fund (No. 17208623), and HKU Seed Fund for Basic Research for New Staff. H.C.S. was funded in part by the Croucher Senior Research Fellowship from Croucher Foundation, and the Health@InnoHK program of the Innovation and Technology Commission under the Hong Kong SAR government.

## Author contributions

R.Z. conceived the general idea, designed, and conducted the EPD-related experiments. R.Z. conducted the COMSOL simulation. C.Z. prepared the superhydrophobic surface, fabricated the EWOD device and conducted EWOD comparison experiments with R.Z. X.F. cultured the living A549 and THP-1 cells and conducted the cell-related experiments. C.C.K.A.Y. provided the fluorescent protein and performed the fluorescent imaging. H.Y.C.L. and R.Z. contributed to the schematic diagram drawing. R.Z. drafted the manuscript, and C.Z., X.F., C.C.K.A.Y., H.Y.C.L., H.L., and H.C.S. provided feedback. H.C.S. and H.L. supervised the project.

## Competing interests

The authors declare the following competing interests: H.C.S. is a scientific advisor of EN Technology Limited, MicroDiagnostics Limited, PharmaEase Tech Limited, and Upgrade Biopolymers Limited in which he owns some equity, and also a managing director of the research center, namely Advanced Biomedical Instrumentation Centre Limited. The works in the paper are however not directly related to the works of these entities. The authors declare no other competing interests.
