## [Peer Review File · Nature Communications]

REVIEWER COMMENTS

Reviewer #1 (Remarks to the Author):

Summary

This paper describes the phenomenon of controlling droplets of different biochemical properties using the principle of electret induced polarization (EPD), which creates a non-uniform electrostatic field through its intrinsic charge. The authors theoretically and experimentally demonstrated the control of droplets of various chemical properties by utilizing the EPD phenomenon. In particular, two driving mechanisms, droplet net charge-induced movement and droplet polarization-induced movement, were proposed for driving EPD, and the effects of the forces applied to the droplets by each driving mechanism were compared and verified experimentally and numerically. To verify the advantage of controlling droplets of various properties compared to EWOD principle, they successfully demonstrated various biochemical droplet control experiments, including conductive/non-conductive droplets and organic/non-organic droplets. Finally, to automate the proposed droplet manipulation, authors designed and built an electromagnetic field induction device that was previously used for electromagnetic droplet control using micro-magnetic particles/beads, and experimentally verified the control of droplets. The reviewer believes that this work can provide useful data and ideas for future microfluidic control technology and design, and recommend publication if the following comments are addressed.

Comments:

- 1) Compared to EWOD, the proposed method has the advantage of controlling various liquids, but judging from the video and results presented, the speed and controllability are significantly lower. What are some ways to overcome this or alternatives to improve it?
- 2) Can the proposed technology operate in air other than oil, and can it operate on a hydrophilic surface in the same environment as the hydrophobic surface?
- 3) The current intrinsic charged electret design makes it difficult to downsize the entire system. It seems that this part of the design needs to be greatly improved in order to become a microfluidic control device. What are alternatives to improve it?

Reviewer #2 (Remarks to the Author):

The authors report a droplet robotic mechanism of EPD, utilizing the polarization of dielectrics in non-uniform electrostatic field. In order to enable automation, they design a multiphysics system to couple the programmable control matrix, grippers and target droplets with the EPD. They demonstrate a number of example cases such as, (i) in-situ image processing function form a “result analysis module” and (ii) temperature control function integrated to form a “heating and cooling module”. The also claim to integrate the efficacies of big data, AI and ML into such system. The point out the applicability of such developments towards the development of high-throughput systems, biomedical diagnostics, materials synthesis, or the handling of the hazardous chemicals. Overall, the work has been performed with a lot of rigor, however, the work lacks novelty because of following reasons:

(i) The science part of this work is novel. Whatever principles have been employed are already available in the standard literature and the authors exploit all the effects very well to show some of the applicabilities.

(ii) The application part of the work is quite exciting wherein complex motions and applications have been achieved, however, that forces the decision towards the submission into a more specific engineering journal than Nature Communications.

(iii) The figure qualities are rather below standard with the patches of brilliance. Major parts of the figures are not visible with significant magnification. The authors should check the matter with a printout wherein many things are either not visible or readable.

(iv) The writeup also suffers heavily because the manuscript has been written in the form of a report while the scientific and technological aspects have taken a back seat. The abstract and conclusions do not highlight what exactly is the target and achievement of/from this work. There is no quantitative target and output. The authors also do not compare and contrast with the existing literature to establish the same.

In view of the above, I am sorry to reject this wonderful piece of work and wish the best to the authors.

Reviewer #3 (Remarks to the Author):

Major Revision

The manuscript "A Droplet Robotic System Enabled by Electret-induced Polarization on Droplet" presents a robotic system for droplet control and manipulation based on the polarization of dielectrics in a non-uniform electrostatic field. The authors have compared the proposed EPD system with other techniques for droplet manipulation and highlighted its advantages for different operable liquid types. They have also proposed a programmable control matrix and multi-physical grippers coupled with an EPD mechanism for full automation. The article is well structured and balances the fundamental aspects. However, the work lacks some technical details. The following major suggestions are to be addressed to improve the quality of the manuscript.

1. EWOD has only been tested in a sandwich droplet configuration in open air and is being compared to EPD in oil surroundings. However, there is no comparison made of EPD with EWOD in an oil environment. Please discuss or compare. Oil surrounding significantly helps avoid hysteresis, reduces biofouling, and improves droplet motion. Authors can refer to the existing literature works related to oil-based EWOD, e.g., *Evaporation and Electrowetting of Sessile Droplets on Slippery Liquid-Like Surfaces and Slippery Liquid-Infused Porous Surfaces (SLIPS)* | *Langmuir* (acs.org); *Axisymmetric and Nonaxisymmetric Oscillations of Sessile Compound Droplets in an Open Digital Microfluidic Platform* | *Langmuir* (acs.org);

2. On Page 5, Line 99 states that 'Unlike without changing any parameter setting.' Ideally, the polarization of different liquid types should depend on charge parameters. Do the electret charges need to change to polarize various organic liquids? Does the EPD work without changing parameters for all liquids? Please elaborate.

3. In Figure 1D, it is unclear whether the comparison is based on new experiments from this work or previous studies in the literature. A scale bar for the score and details of analysis could be helpful for better readability.

4. Please include a discussion on the limitations of EPD. This can include information on the size of droplets that EPD can handle, the characteristics of the surrounding medium, and other different EPD parameters.

5. In the supplementary videos for the gripper platform, it is not clear whether the droplet is moving around the vicinity of any of the four electrets or is precisely stable at its intended position, which is the centre of the four electrets. Please explain.

6. Please detail the comparison of resolution and precision of EPD compared to other techniques (like EWOD), including some dimensional data, such as the mean resolution and standard deviation, for their ability to move/displace liquid droplets.

7. Important aspects related to the coalescence or merging of droplets in an oil medium have been missed. It would be helpful if the authors could provide some explanation about these aspects to aid the readers' comprehension. Authors can use this literature for references: Manipulation and control of droplets on surfaces in a homogeneous electric field | Nature Communications; Effect of electrowetting induced capillary oscillations on coalescence of compound droplets - ScienceDirect.

8. Please review the manuscript carefully for any grammatical errors or typos and correct them as needed.

Response to the reviewed manuscript:

“A Droplet Robotic System Enabled by Electret-induced Polarization on Droplet”

Reviewer #1

This paper describes the phenomenon of controlling droplets of different biochemical properties using the principle of electret induced polarization (EPD), which creates a non-uniform electrostatic field through its intrinsic charge. The authors theoretically and experimentally demonstrated the control of droplets of various chemical properties by utilizing the EPD phenomenon. In particular, two driving mechanisms, droplet net charge-induced movement and droplet polarization-induced movement, were proposed for driving EPD, and the effects of the forces applied to the droplets by each driving mechanism were compared and verified experimentally and numerically. To verify the advantage of controlling droplets of various properties compared to EWOD principle, they successfully demonstrated various biochemical droplet control experiments, including conductive/non-conductive droplets and organic/non-organic droplets. Finally, to automate the proposed droplet manipulation, authors designed and built an electromagnetic field induction device that was previously used for electromagnetic droplet control using micro-magnetic particles/beads, and experimentally verified the control of droplets. The reviewer believes that this work can provide useful data and ideas for future microfluidic control technology and design, and recommend publication if the following comments are addressed.

Response: We thank Reviewer 1 for the thorough examination of our manuscript and providing insightful comments. The comments are instrumental in guiding the revision and strengthening our revised manuscript. We also appreciate the Reviewer’s positive assessment of the presented technology. As described below and reflected upon in the revised manuscript, we have addressed all of Reviewer’s valuable comments.

Comment 1.1: Compared to EWOD, the proposed method has the advantage of controlling various liquids, but judging from the video and results presented, the speed and controllability are significantly lower. What are some ways to overcome this or alternatives to improve it?

Response: We thank the Reviewer for this valuable comment. To further enhance EPD’s speed and controllability, improvements can be implemented on two different levels: At the actuation system level, the magnetic force generated by the programmable control matrix can be further enhanced by increasing the current (e.g. replacing switch ICs with higher current threshold chips) or modifying the coil design (e.g. more layers of coils), increasing system robustness and switching frequency. At the actuation material level, we can utilize different electret materials (e.g. CYTOP, which can provide a higher surface charge density up to 2 mC/m^2 for a $15\text{-}\mu\text{m}$ thick film ¹) or apply different charging methods (e.g. corona charging or electron-beam irradiation ²) to further increase the surface charge density of the electret, providing a larger actuation force. As a demonstration, we have added theoretical analysis to show that the increased charge density of the electret can

significantly enhance the actuation speed (newly added Supplementary Note S2), consistent with the experimental results (revised Main Fig. 2H). The maximum speed we reported in this work can reach 60 mm/s (revised Main Fig. 3B), comparable with EWOD or other droplet actuation techniques (newly added Supplementary Table S1). In response, we have added a supplementary note, corresponding discussions, and the following figures and table to the revised manuscript (Line 190-193, 203-205, and 514-522).

1. Revised Main Fig. 2H:

Fig. 2. (H) The relationship between the maximum actuation velocity of a 10 μL droplet and the absolute charge amount of electret under different height differences. Error bars, SD.

2. Newly added Supplementary Table S1:

Excerpt of Table S1. Comparison between EPD and other common techniques, including EWOD³⁻¹¹, magnetic¹²⁻²³, acoustic²⁴⁻⁴⁰, and thermal⁴¹⁻⁴⁷ based droplet actuation platform.

	EPD (this work)	EWOD		Magnetic	Acoustic	Thermal
		Air-based	Oil-based			
.....
Speed	High (22-60 mm/s, Fig. 3B)	High (140 mm/s) ¹⁰	High (45 mm/s) ¹⁰	High (3-50 mm/s) ^{12,16,21}	High (10-50 mm/s) ³⁸	Low (0.01-0.3 mm/s) ^{41,46}
.....

Comment 1.2: Can the proposed technology operate in air other than oil, and can it operate on a hydrophilic surface in the same environment as the hydrophobic surface?

Response: We thank the Reviewer for proposing the instructive comment. The proposed EPD can actuate droplet in the environment of air on a hydrophobic surface, as shown in the revised Main Fig. 4g. As for the hydrophilic surface, droplet cannot be actuated on it due to the increasing resistance (newly added Supplementary Fig. S14 and Note S4), similar to the restriction of most planar droplet actuation platforms, including EWOD^{7,48} and magnetic-based platform^{12,13} (newly added Supplementary Table S1). To address this comment, we have added a supplementary note, corresponding discussions, and the following figures and table to the revised manuscript (Line 311-317).

## 1. Revised Main Fig. 4:

Fig. 4. (F) Schematic diagram of EPD's compatibility with various surroundings and substrates. (G) Optical images showing the EPD-based droplet actuation in air, on the hydrophobic surface. Scale bar: 3 mm. (H) Optical images showing the EPD-based actuation of droplet floating at oil-air interface, with an oil substrate. Scale bar: 3 mm. (I) Optical images showing the EPD-based actuation of droplet merged in oil surrounding, with a hydrophobic substrate. Scale bar: 3 mm.

## 2. Newly added Supplementary Fig. S14:

Figure S14. The EPD-based droplet actuation on hydrophilic surface. The droplet exhibits a tendency to move with different contact angles on advancing and receding sides. Compared with the hydrophobic surface, the increasing resistance on the hydrophilic surface leads to the failure in droplet actuation.

## 3. Newly added Supplementary Table S1:

Excerpt of Table S1. Comparison between EPD and other common techniques, including EWOD[3-11], magnetic[12-23], acoustic[24-40], and thermal[41-47] based droplet actuation platform.

| | | EPD (this work) | EWOD | | Magnetic | Acoustic | Thermal |
|---|---|---|---|---|---|---|---|
| | | | Air-based | Oil-based | | | |
| … | … | … | … | | … | … | … |
| Compatibility with surroundings /substrates | Droplet surrounding medium | High (oil/air/oil-air interface, Fig. 4) | High (air/oil) | | High (oil/air) [13] | High (air/oil) [24,25] | Low (air only) [41,43] |
| | Substrate surface | Medium (hydrophobic/ oil substrate, Fig. 4) | Medium (hydrophobic) [7] | | Medium (hydrophobic) [13] | High (hydrophobic/ hydrophilic) [36] | Low (hydrophilic surface with hydrophobic pattern) [41,43] |
| | Substrate material | High (no demand) | Low (limited by thickness and permittivity) [8,9] | | Medium (some require flexible substrate impregnated with magnetic materials) [21,22] | Low (piezoelectric materials) [25,37] | Medium (thickness and thermal properties of the substrate affect heating efficiency) [43] |
| … | … | … | … | | … | … | … |

Comment 1.3: The current intrinsic charged electret design makes it difficult to downsize the entire system. It seems that this part of the design needs to be greatly improved in order to become a microfluidic control device. What are alternatives to improve it?

Response: We thank the Reviewer for outlining this instructive point. The size of the system depends on the size of target droplets in the specific application scenario. The system we show in the paper is mainly designed to manipulate droplets on a μL scale. If we downsize the system while maintaining the original design, the charge density of the electret needs to be increased to ensure functionality (newly added Supplementary Fig. S28). Besides, we can also microfabricate a 2D charge distribution-controllable electret material. This approach has the potential to actuate droplets by directly programming the local electric field distribution without physically moving the electret material, thus further reducing the size of the system. This is an important topic that deserves a separate thorough investigation currently being conducted in another work. In response to Reviewer's comment, we have added corresponding discussions and the following figure to the revised manuscript (Line 507-514).

Figure S28. EPD force exerted by a downsized EPD gripper calculated based on the simulation result. If the EPD gripper ($12 \times 12 \times 8$ mm) is downsized by 10 times to about $1.2 \times 1.2 \times 0.8$ mm, the fabrication of the electret materials is still achievable in the micromachining process. Based on the simulation result, the EPD force exerted by the smaller gripper on the same-volume droplet is also reduced, where the maximum value of the EPD force is reduced by about 26.6 times. To generate a EPD force comparable to the original one, the charge density of the gripper, ϕ , needs to be increased to about -8.25×10^{-5} C/m². The y-coordinate of the gripper is set as 0 mm, and the component of the Maxwell force in the x-direction is calculated and compared.

Reviewer #2

The authors report a droplet robotic mechanism of EPD, utilizing the polarization of dielectrics in non-uniform electrostatic field. In order to enable automation, they design a multiphysics system to couple the programmable control matrix, grippers and target droplets with the EPD. They demonstrate a number of example cases such as, (i) in-situ image processing function form a “result analysis module” and (ii) temperature control function integrated to form a “heating and cooling module”. They also claim to integrate the efficacies of big data, AI and ML into such system. They point out the applicability of such developments towards the development of high-throughput systems, biomedical diagnostics, materials synthesis, or the handling of the hazardous chemicals. Overall, the work has been performed with a lot of rigor, however, the work lacks novelty because of following reasons:

Response: Thank you for summarizing our paper and for highlighting potential concerns regarding the novelty of our work. We truly value your feedback and appreciate your input.

Before we elaborate on the novelty of this work, we would like to clarify that the summary being made above: “They demonstrate a number of handling of the hazardous chemicals” is mostly based on the discussion regarding the potential enhancements and future implementations of the EPD-based robotic system. They are not yet “demonstrated” within the current work but are proposed as part of our vision for how the system can evolve in the future. We apologize for the confusion we may have caused. As a result, we have revised the discussion section of our manuscript, clearly indicating that those points are future directions.

In this study, we focus on the investigation of the newly proposed droplet actuation mechanism, termed electret-induced polarization on droplet (EPD), and its corresponding implementation in an automated droplet robotic system. The novelty of our work lies in both mechanism and application, including the unique EPD mechanism, superior properties, and the design of a fully automated system. The response to comment 2.1 and 2.2 will explain the scientific novelty and novelty of applications in detail.

We have also realized that our writing may not have clearly highlighted the novelty, thus we have revised our writing flow throughout the paper, especially in the Abstract, Introduction, and Discussion sections. In these revised sections, we now emphasize the novelty and quantitative achievement of this paper, and systematically compare the differences between our work and other existing techniques. Detailed revisions will be elaborated in the response to comment 2.4.

Comment 2.1: The science part of this work is novel. Whatever principles have been employed are already available in the standard literature and the authors exploit all the effects very well to show some of the applicabilities.

Response: We thank the Reviewer for assessing the novelty of our work. Here, we would like to further clarify our scientific novelty from the following aspects:

- **Novel droplet moving phenomenon enabled by electret's attraction:** In our study, the proposed droplet robotic system is enabled by a newly observed attraction effect between various liquid droplets and the intrinsically charged electret. We find that electrets, either positively or negatively charged, can attract a wide range of liquid droplets, including inorganic/organic and conductive/dielectric liquids with relative permittivity ranging from 2.25 to 84.2 and volume from 500 nL to 1 mL, different from the previously reported attraction/repulsion phenomena induced by the net charge of droplets⁴⁹⁻⁵².
- **Unique mechanism of “electrostatic charge-based” liquid polarization:** Through systematic simulation and experimental investigation, the newly proposed droplet actuation mechanism is validated as the electret-induced polarization on droplet (EPD): With the intrinsic charges, electret generates a non-uniform electrostatic field, polarizing and attracting droplets.

The most fundamental physical principle of EPD can be attributed to polarization, a principle that is indeed already present and discussed in the literature about micro-object actuation. However, most studies of polarization-based actuation are targeted at manipulating solid particles or cells^{53,54}. The investigation on the polarization effect in liquids has been quite limited so far, with a few studies examining the influence of high-frequency and/or high-voltage AC/DC electric fields on liquid polarization⁵⁵⁻⁵⁸. Up to now, there has been no systematic study of the liquid polarization generated by a material carrying intrinsic electrostatic charges, not to mention the implementation of automated droplet actuation based on this mechanism. According to the analysis presented in this work (newly added Supplementary Note S1), the polarizations induced by AC/DC and electrostatic charges differ at the level of equivalent circuit models, leading to different variables and application scenarios. Therefore, our study of the EPD mechanism can add some unique scientific understandings by complementing the existing principle of liquid polarization from electrostatic perspective, while also inspiring diverse droplet actuation functionalities.

- **Unprecedented properties brought by EPD mechanism:** Benefiting from the novel EPD mechanism, we have validated the droplet actuation with unprecedented applicability to a wide range of liquids and biochemical samples, both theoretically and experimentally. EPD's polarization-based principle allows it to actuate various inorganic/organic liquids, while its unique nature of utilizing electrostatic charges further avoids the undesired effects of electric field frequency and liquid conductivity (newly added Supplementary Note S1), thus enabling all liquid actuation in principle. The use of electrostatic charges in EPD can also eliminate Joule heating and dielectric loss heating^{57,59-62}, improving its biocompatibility. Therefore, these advantages brought by the EPD mechanism effectively address the challenges commonly faced by current liquid manipulation technologies.

Thanks to the Reviewer's comment, we recognize that it would be helpful to highlight the scientific novelty of the EPD mechanism in the manuscript. To address the Reviewer's comment, we have added a Supplementary Note S1 and revised the Main Fig. 2 with the corresponding descriptions in the manuscript to clearly state the novelty of our EPD mechanism (Line 81-83, 92, 98-99, 136-139, 228-231, 482-490).

Comment 2.2: The application part of the work is quite exciting wherein complex motions and applications have been achieved, however, that forces the decision towards the submission into a more specific engineering journal than Nature Communications.

Response: Thank you for recognizing the application part of this work as an exciting demonstration. We would like to clarify that our work does not just focus on solving a specific engineering problem, but instead, introducing a novel droplet actuation technology which encompasses a coherent and comprehensive study from the EPD phenomenon, mechanism, and properties (as discussed in the Response 2.1) to its potential applications. Among them, the demonstrated application is only an example of the automated implementations of the proposed droplet actuation technology. The actual implication of the proposed droplet actuation technology is much broader than the specific engineering issues involved in the demonstrated application examples.

To further display the implication of our EPD system on multidisciplinary liquid-based applications, in the revised manuscript, we have included additional experiments as a demonstration (Newly added Supplementary Fig. S27). Benefiting from EPD's excellent biocompatibility, liquid generality, and automation, here, we utilize the EPD system to explore the dynamic generation of inflammatory mediator based on in vitro cell-bacteria models. Our result reveals the non-monotonic relationship between the IL-1 β concentration and repeated bacterial infections, which helps to explore the generation of inflammatory mediators and the connection between diseases and inflammatory mediators⁶³⁻⁶⁵. Combined with the original application example of automating a bioassay for diverse biofluids (Fig. 6), our demonstrations validate the wide applicability and impact of our EPD-based droplet robotic system on multidisciplinary scientific research that requires precise liquid manipulations, including clinical, biological, and engineering sciences.

We appreciate the Reviewer's point about the need to expand the implications of our work beyond engineering, which leads to significant improvements of our paper. We have revised the Result section (Line 462-479) of our manuscript and added the following figure to further demonstrate the implication of the EPD-based system on scientific research in multiple fields. Given the cross-disciplinary nature of our study, we believe that it fits well for publication in Nature Communications, "a multidisciplinary journal aimed at serving audience with diverse backgrounds in biological, health, physical, chemical, ..., applied, and engineering sciences."⁶⁶

Figure S27. The application of the EPD-based droplet robotic system for establishing in vitro cell-bacteria model of inflammation and in-situ detecting inflammatory mediator of human IL-1 β . (A) Schematic diagram of the workflow of establishing cell-bacteria model of inflammation and in-situ detecting the concentration of inflammatory mediator on EPD-based system. Scale bar: 50 μ m. (B) Standard curve of the concentration of human IL-1 β . (C) The detected concentration of human IL-1 β after 1st, 2nd, and 3rd infection of bacteria. Error bars, SD (n = 3).

Comment 2.3: The figure qualities are rather below standard with the patches of brilliance. Major parts of the figures are not visible with significant magnification. The authors should check the matter with a printout wherein many things are either not visible or readable.

Response: We thank the Reviewer for this valuable comment. We have increased the resolution of all the images in the manuscript. However, considering that the submission system will automatically convert the manuscript into PDF format and compress the file, we have also provided all the figures in the form of “.jpg” in the "Supplementary Files-Original Figures" for Reviewer’s convenience.

Comment 2.4: The writeup also suffers heavily because the manuscript has been written in the form of a report while the scientific and technological aspects have taken a back seat. The abstract and conclusions do not highlight what exactly is the target and achievement of/from this work. There is no quantitative target and output. The authors also do not compare and contrast with the existing literature to establish the same.

Response: We thank the Reviewer for assessing the writeup of our manuscript. To address your comment, we have made several significant revision and improved the whole manuscript, including: 1) highlighting the quantitative goals and achievements in the Abstract, Introduction, and Discussion sections (Line 27-30, 92-97, 102-105, 491-504); 2) analyzing the difference between EPD and other existing mechanisms, and emphasizing the scientific novelty of our work (Note S1, Line 81-83, 92, 98-99, 136-139, 228-231, 482-490); 3) adding extensive experiments (including Fig. S2, 3, 8, 10, 11, and 21) to compare EPD with other existing droplet-actuation technologies and summarizing as a Supplementary Table (Table S1, as shown below); 4) raising the

existing limitations of EPD and discussing its potential future development directions (Line 312-317, 389-391, 507-522). All the revisions are blue-highlighted in the revised manuscript for the Reviewer’s convenience.

Table S1. Comparison between EPD and other common techniques, including EWOD³⁻¹¹, magnetic¹²⁻²³, acoustic²⁴⁻⁴⁰, and thermal⁴¹⁻⁴⁷ based droplet actuation platform.

		EPD (this work)	EWOD		Magnetic	Acoustic	Thermal
			Air-based	Oil-based			
Generality with operable liquids	Droplet size	High (over 500 nL-1 mL, Fig. 3B)	Low (depends on electrode size, nL-10 μ L) ³⁻⁵		Medium (1 μ L-200 μ L) ^{12,13}	Medium (200 nL-50 μ L) ²⁴⁻²⁶	Low (depends on microheater size, 3 nL-1 μ L) ⁴¹⁻⁴³
	Liquid types	High (Fig 3)	Low (high conductive/permittivity, Fig. S2)	Low (high conductive/permittivity, Fig. S2)	Medium (prefer high surface tension & low viscosity) ^{12,13}	High ^{26,27}	Medium (non-volatile liquid) ⁴¹
	Other requirement	No	No	Immiscible with silicone oil ⁶	Not catalyzed/reactive with magnetic nanoparticles ^{14,15}	No acids or bases on ZnO substrate ²⁵	Prefer liquid with highly temperature-dependent surface tension ⁴¹
Compatibility with bio-samples	Body fluids	High (Fig.1D)	Low (Fig.S3)	Low (Fig.S3, S8)	High ^{16,17}	High ^{28,29}	Medium (heating may affect activity of biomarker inside) ⁴⁴
	Protein	High (Fig.4C)	Low (Fig. S9)	Medium (Fig. S10)	High ¹⁸	Medium (acoustic heating may affect activity) ^{25,30,31}	Low (direct heating may damage activity) ⁴⁵
	Living cells	High (Fig.4D,E)	Low (Fig. 4E)	Low (Fig. S11)	High ¹⁹	Medium (increased amplitude/decreased frequency may break cell membranes) ³²⁻³⁵	Low (direct heating may damage activity) ⁴⁵
	Bioassay process	High (Fig. 6)	Low (biofouling) ⁶	High	Low (particle’s peroxidase-like activity & non-transparency) ^{14,15,20}	Low (acoustic heating affects temperature-dependent detection) ^{25,31}	Low (cross-contamination) ⁴³
Compatibility with surroundings /substrates	Droplet surrounding medium	High (oil/air/oil-air interface, Fig. 4)	High (air/oil)		High (oil/air) ¹³	High (air/oil) ^{24,25}	Low (air only) ^{41,43}
	Substrate surface	Medium (hydrophobic/ oil substrate, Fig. 4)	Medium (hydrophobic) ⁷		Medium (hydrophobic) ¹³	High (hydrophobic/hydrophilic) ³⁶	Low (hydrophilic surface with hydrophobic pattern) ^{41,43}
	Substrate material	High (no demand)	Low (limited by thickness and permittivity) ^{8,9}		Medium (some require flexible substrate impregnated with magnetic materials) ^{21,22}	Low (piezoelectric materials) ^{25,37}	Medium (thickness and thermal properties of the substrate affect heating efficiency) ⁴³
Speed	High (22-60 mm/s, Fig. 3B)	High (140 mm/s) ¹⁰	High (45 mm/s) ¹⁰	High (3-50 mm/s) ^{12,16,21}	High (10-50 mm/s) ³⁸	Low (0.01-0.3 mm/s) ^{41,46}	
Resolution	Depends on coil size, e.g., 1.5 mm (Fig. S21)	Depends on electrode size, e.g., 2 mm (Fig. S21)		Depends on electromagnet size, e.g., 1.9 mm (Fig. S21)	Depends on droplet volume and RF pulse duration, e.g., 1mm ³⁶	Depends on microheater size, e.g., 1mm ^{41,43}	
Simplicity of programming	High (switch matrix, Fig. 5)	High (switch matrix) ¹¹		High (switch matrix/translation stage/alternating electromagnetic field) ^{13,23}	Low (modulate frequency, amplitude, and phase difference of one/multiple actuators) ^{39,40}	Medium (switch matrix, with an optional temperature feedback control) ^{43,47}	
Working Voltage	Low (5.5V)	High (60 V-150 V, depends on liquid types, Fig. S2, S3, S8)		Medium (several-tens of volts) ^{12,13,16,23}	Medium (40 V-70 V) ²⁵	Medium (2.5V-25V) ^{41,43,46}	
Cost of fabrication	Low (consumables: lower than US\$1; control system: ~US\$100)	Medium (consumables: US\$42.6; control system: ~US\$500)		Medium (consumables: lower than US\$1; control system: ~US\$100-US\$1000) ^{12,13,16,23}	High (consumables: ~US\$70; control system: ~US\$600-US\$3000) ^{39,40}	Medium (consumables: ~US\$10; control system: ~US\$100-US\$1000) ^{41,43,47}	

Comment 2.5: In view of the above, I am sorry to reject this wonderful piece of work and wish the best to the authors.

Response: We truly appreciate the Reviewer's effort in assessing and summarizing our work. Thanks to your comments above, we were able to realize that our shortcomings in the writing and figure qualities have limited the demonstration of our novelty. We have thus further explored the novelty of our work in both science and application aspects and revised the manuscript accordingly to highlight it. We believe that our revisions and responses have addressed all the comments and significantly improved our manuscript.

Reviewer #3

Major Revision.

The manuscript "A Droplet Robotic System Enabled by Electret-induced Polarization on Droplet" presents a robotic system for droplet control and manipulation based on the polarization of dielectrics in a non-uniform electrostatic field. The authors have compared the proposed EPD system with other techniques for droplet manipulation and highlighted its advantages for different operable liquid types. They have also proposed a programmable control matrix and multi-physical grippers coupled with an EPD mechanism for full automation. The article is well structured and balances the fundamental aspects. However, the work lacks some technical details. The following major suggestions are to be addressed to improve the quality of the manuscript.

Response: We thank Reviewer 3 for the thorough, positive, and encouraging review of our manuscript and providing constructive feedback, from which we have benefited a great deal in preparing the revised manuscript. Following the Reviewer's comments, we have performed additional experiments and added additional data and clarifying points in the revised manuscript. All the Reviewer's comments have been addressed in a point-by-point manner to provide more technical details as expanded upon below.

Comment 3.1: EWOD has only been tested in a sandwich droplet configuration in open air and is being compared to EPD in oil surroundings. However, there is no comparison made of EPD with EWOD in an oil environment. Please discuss or compare. Oil surrounding significantly helps avoid hysteresis, reduces biofouling, and improves droplet motion. Authors can refer to the existing literature works related to oil-based EWOD, e.g., Evaporation and Electrowetting of Sessile Droplets on Slippery Liquid-Like Surfaces and Slippery Liquid-Infused Porous Surfaces (SLIPS) | *Langmuir* (acs.org); Axisymmetric and Nonaxisymmetric Oscillations of Sessile Compound Droplets in an Open Digital Microfluidic Platform | *Langmuir* (acs.org);

Response: We thank the Reviewer for the valuable comment, and we have included extra experiments to compare EPD with EWOD in an oil environment.

As the Reviewer pointed out, the oil environment does decrease the contact line friction¹⁰, reduce the protein adsorption and biofouling⁶⁷. However, compared with EPD, the oil-based EWOD still has a significantly lower range of operable liquid types (newly added Supplementary Fig. S2) and lower compatibility with bio-samples, including body fluids (newly added Supplementary Fig. S3 and S8), protein solution (Main Fig. 4C & newly added Supplementary Fig. S10), and living cells (newly added Supplementary Fig. S11). To address this comment, we have added the following figures with descriptions to the revised manuscript and systematically summarized our results in the newly added Supplementary Table S1. We have also added the above-mentioned literature as references to the revised manuscript.

1. Newly added Supplementary Fig. S2:

Figure S2. Comparison between EPD and EWOD's operable liquid types by demonstrating the actuation of various inorganic/organic liquids. The demonstrated liquid types include (A) water (inorganic), (B) glycerol (alcohol), and (C) triacetin (ester). The position of the front edge of the droplet normalized by the distance between two neighboring coils/electrodes is defined as the relative moving distance. The result indicates that EWOD faces challenges when actuating organic liquids with low dielectric constants, even when oil surrounding is introduced. In contrast, EPD can actuate them smoothly without changing any of the parameter settings, showing EPD's generality of various liquids.

2. Newly added Supplementary Fig. S3:

Figure S3. Comparison between EPD and EWOD's compatibility with bio-samples by demonstrating the actuation of various body fluids. The demonstrated body fluids include (A) serum, (B) saliva, and (C) urine. The position of the front edge of the droplet normalized by the distance between two neighboring coils/electrodes is defined as the relative moving distance. The result indicates that air-based EWOD faces challenges when actuating biological fluids. When silicone oil environment is introduced to EWOD, the actuation of three body fluids gets improved to varying degrees. However, all three body fluids experience droplet fragmentation during actuation due to the high viscous resistance induced by the oil environment, which prevents subsequent continuous actuation¹⁰. In contrast, EPD can actuate them smoothly without changing any of the parameter settings, showing EPD's compatibility with body fluids.

3. Newly added Supplementary Fig. S8:

Figure S8. Actuation of various body fluids on oil-based EWOD. (A) Maximum actuation voltages of serum, saliva, and urine on oil-based EWOD without droplet fragmentation. The result shows that the critical voltages of the three body fluids vary; therefore, the setting of actuation voltage needs to be customized according to the type of body fluids. (B) Actuation of human serum, saliva, and urine on oil-based EWOD with a working voltage of 65 V (highest voltage preventing all three body fluids from fragmentation). The position of the front edge of the droplet normalized by the distance between two neighboring electrodes is defined as the relative moving distance. The result demonstrates that the tested oil-based EWOD still faces challenges when actuating body fluids with a higher concentration of protein, while the actuation performance varies significantly among different biofluids.

4. Newly added Supplementary Fig. S10:

Figure S10. Actuation of protein solution on oil-based EWOD. (A) Actuation of 0.05 mg/mL FITC-BSA on oil-based EWOD with working voltage of 100 V, in which droplet fragmentation happens and prevents subsequent continuous actuation. (B) Fluorescence microscope images showing the protein adsorption on oil-based EWOD after actuating a droplet of 0.05 mg/mL FITC-BSA solution, indicated by the change of fluorescence intensity on the substrate.

5. Newly added Supplementary Fig. S11:

Figure S11. Comparison of the compatibility with living cells of EPD and oil-based EWOD. (A) Impact of EPD effect and electrowetting effect on cell activity, indicated by the variation of the percentage of living cells after being treated on EPD/ oil-based EWOD for 30 mins and then incubated for 12 hours. The activity of cells treated by oil-based EWOD decreases with a higher degree than that of cells treated by EPD, suggesting that the introduction of oil environment still could not completely avoid the effect of Joule heating and high electric field strength on cell activity. Error bars, SD (n = 3). (B) Impact of EPD and oil-based EWOD system setup on cell culture, indicated by the variation of the percentage of living cells after being cultured for 24 hours on EPD/oil-based EWOD without power supply. The percentage of living cells cultured on the oil-based EWOD (58.5%) is significantly lower than that of cells cultured on the EPD (91.2%). This is likely because the silicone oil environment on the double-plate EWOD limits the gas exchange between the droplet and the atmosphere, thus leading to asphyxiation of the cells within the droplet. Error bars, SD (n = 3).

## 6. Newly added Supplementary Table S1:

Excerpt of Table S1. Comparison between EPD and other common techniques, including EWOD[3-11], magnetic[12-23], acoustic[24-40], and thermal[41-47] based droplet actuation platform.

| | | EPD (this work) | EWOD | | Magnetic | Acoustic | Thermal |
|---|---|---|---|---|---|---|---|
| | | | Air-based | Oil-based | | | |
| … | … | … | … | | … | … | … |
| Generality with operable liquids | Droplet size | High (over 500 nL-1 mL, Fig. 3B) | Low (depends on electrode size, nL-10 μL) [3-5] | | Medium (1 μL-200 μL) [12,13] | Medium (200 nL-50 μL) [24-26] | Low (depends on microheater size, 3 nL-1 μL) [41-43] |
| | Liquid types | High (Fig 3) | Low (high conductive/ permittivity, Fig. S2) | Low (high conductive/ permittivity, Fig. S2) | Medium (prefer high surface tension & low viscosity) [12,13] | High [26,27] | Medium (non-volatile liquid) [41] |
| | Other requirement | No | No | Immiscible with silicone oil [6] | Not catalyzed/reactive with magnetic nanoparticles [14,15] | No acids or bases on ZnO substrate [25] | Prefer liquid with highly temperature-dependent surface tension [41] |
| Compatibility with bio-samples | Body fluids | High (Fig.1D) | Low (Fig.S3) | Low (Fig.S3, S8) | High [16,17] | High [28,29] | Medium (heating may affect activity of biomarker inside) [44] |
| | Protein | High (Fig.4C) | Low (Fig. S9) | Medium (Fig. S10) | High [18] | Medium (acoustic heating may affect activity) [25,30,31] | Low (direct heating may damage activity) [45] |
| | Living cells | High (Fig.4D,E) | Low (Fig. 4E) | Low (Fig. S11) | High [19] | Medium (increased amplitude/ decreased frequency may break cell membranes) [32-35] | Low (direct heating may damage activity) [45] |
| | Bioassay process | High (Fig. 6) | Low (biofouling) [6] | High | Low (particle's peroxidase-like activity & non-transparency) [14,15,20] | Low (acoustic heating affects temperature-dependent detection) [25,31] | Low (cross-contamination) [43] |
| … | … | … | … | … | … | … | … |

Comment 3.2: On Page 5, Line 99 states that ‘Unlike without changing any parameter setting.’ Ideally, the polarization of different liquid types should depend on charge parameters. Do the electret charges need to change to polarize various organic liquids? Does the EPD work without changing parameters for all liquids? Please elaborate.

Response: We thank the Reviewer for this valuable comment. The electret charge does not need to be changed to polarize various organic liquids (the charge density we commonly use in experiments is $-1.74 \times 10^{-5} \text{ C/m}^2$). The EPD system can also work without changing parameters for all liquids we demonstrated. The reason why the parameters need no changing is that the EPD forces generated based on the existing experimental conditions are sufficient to actuate most liquid types. The force generated by EPD decreases as the liquid droplet’s permittivity decreases, as shown in Main Fig. 3D. However, the EPD force (calculated based on the surface charge density of $-1.74 \times 10^{-5} \text{ C/m}^2$) for liquid with the minimum permittivity is still much larger than the minimum force required for effective actuation (Main Fig. 3D&F). Therefore, in our experiments, without adjusting parameters, such as electret charge density, size of the EPD gripper, etc., water/glycerol/hexadecane can still be actuated despite the significant difference in their permittivities (Response Fig. R1). To address this comment, we have added some explanations in the revised manuscript (Line 241-243).

Figure R1. Actuation of various inorganic/organic liquids with different permittivities on EPD-based droplet robotic system.

Comment 3.3: In Figure 1D, it is unclear whether the comparison is based on new experiments from this work or previous studies in the literature. A scale bar for the score and details of analysis could be helpful for better readability.

Response: We thank the Reviewer for proposing the instructive comment. To address this comment, a scale bar for the evaluation result is added to the revised Main Fig. 1D, and the evaluation rules are explained in the caption and Methods section. The details of analysis are included in the newly added Supplementary Table S1,

which is either based on the results of our experiments (labeled with the number of the figure in which relative data are presented) or based on the references (labeled with the number of the specific reference).

1. Revised Main Fig. 1D:

Fig. 1. (D) Comparison between the proposed EPD-based droplet robotic system and the existing automated droplet actuation system from 6 perspectives (left). The evaluations are conducted based on the detailed information in Supplementary Table S1. EPD's generality with operable liquid types and the compatibility with bio-sample are particularly demonstrated and compared with air-based EWOD and oil-based EWOD based on fig S2 and S3, respectively (right).

2. Newly added Supplementary Table S1:

Table S1. Comparison between EPD and other common techniques, including EWOD³⁻¹¹, magnetic¹²⁻²³, acoustic²⁴⁻⁴⁰, and thermal⁴¹⁻⁴⁷ based droplet actuation platform.

		EPD (this work)	EWOD		Magnetic	Acoustic	Thermal
			Air-based	Oil-based			
Generality with operable liquids	Droplet size	High (over 500 nL-1 mL, Fig. 3B)	Low (depends on electrode size, nL-10 μ L) ³⁻⁵		Medium (1 μ L-200 μ L) ^{12,13}	Medium (200 nL-50 μ L) ²⁴⁻²⁶	Low (depends on microheater size, 3 nL-1 μ L) ⁴¹⁻⁴³
	Liquid types	High (Fig 3)	Low (high conductive/permittivity, Fig. S2)	Low (high conductive/permittivity, Fig. S2)	Medium (prefer high surface tension & low viscosity) ^{12,13}	High ^{26,27}	Medium (non-volatile liquid) ⁴¹
	Other requirement	No	No	Immiscible with silicone oil ⁶	Not catalyzed/reactive with magnetic nanoparticles ^{14,15}	No acids or bases on ZnO substrate ²⁵	Prefer liquid with highly temperature-dependent surface tension ⁴¹
Compatibility with bio-samples	Body fluids	High (Fig.1D)	Low (Fig.S3)	Low (Fig.S3, S8)	High ^{16,17}	High ^{28,29}	Medium (heating may affect activity of biomarker inside) ⁴⁴
	Protein	High (Fig.4C)	Low (Fig. S9)	Medium (Fig. S10)	High ¹⁸	Medium (acoustic heating may affect activity) ^{25,30,31}	Low (direct heating may damage activity) ⁴⁵
	Living cells	High (Fig.4D,E)	Low (Fig. 4E)	Low (Fig. S11)	High ¹⁹	Medium (increased amplitude/ decreased frequency may break cell membranes) ³²⁻³⁵	Low (direct heating may damage activity) ⁴⁵
	Bioassay process	High (Fig. 6)	Low (biofouling) ⁶	High	Low (particle's peroxidase-like activity & non-transparency) ^{14,15,20}	Low (acoustic heating affects temperature-dependent detection) ^{25,31}	Low (cross-contamination) ⁴³
Compatibility with surroundings /substrates	Droplet surrounding medium	High (oil/air/oil-air interface, Fig. 4)	High (air/oil)		High (oil/air) ¹³	High (air/oil) ^{24,25}	Low (air only) ^{41,43}
	Substrate surface	Medium (hydrophobic/ oil substrate, Fig. 4)	Medium (hydrophobic) ⁷		Medium (hydrophobic) ¹³	High (hydrophobic/ hydrophilic) ³⁶	Low (hydrophilic surface with hydrophobic pattern) ^{41,43}
	Substrate material	High (no demand)	Low (limited by thickness and permittivity) ^{8,9}		Medium (some require flexible substrate impregnated with magnetic materials) ^{21,22}	Low (piezoelectric materials) ^{25,37}	Medium (thickness and thermal properties of the substrate affect heating efficiency) ⁴³
Speed	High (22-60 mm/s, Fig. 3B)	High (140 mm/s) ¹⁰	High (45 mm/s) ¹⁰	High (3-50 mm/s) ^{12,16,21}	High (10-50 mm/s) ³⁸	Low (0.01-0.3 mm/s) ^{41,46}	
Resolution	Depends on coil size, e.g., 1.5 mm (Fig. S21)	Depends on electrode size, e.g., 2 mm (Fig. S21)		Depends on electromagnet size, e.g., 1.9 mm (Fig. S21)	Depends on droplet volume and RF pulse duration, e.g., 1mm ³⁶	Depends on microheater size, e.g., 1mm ^{41,43}	
Simplicity of programming	High (switch matrix, Fig. 5)	High (switch matrix) ¹¹		High (switch matrix/translation stage/alternating electromagnetic field) ^{13,23}	Low (modulate frequency, amplitude, and phase difference of one/multiple actuators) ^{39,40}	Medium (switch matrix, with an optional temperature feedback control) ^{43,47}	
Working Voltage	Low (5.5V)	High (60 V-150 V, depends on liquid types, Fig. S2, S3, S8)		Medium (several-tens of volts) ^{12,13,16,23}	Medium (40 V-70 V) ²⁵	Medium (2.5V-25V) ^{41,43,46}	
Cost of fabrication	Low (consumables: lower than US\$1; control system: ~US\$100)	Medium (consumables: US\$42.6; control system: ~US\$500)		Medium (consumables: lower than US\$1; control system: ~US\$100-US\$1000) ^{12,13,16,23}	High (consumables: ~US\$70; control system: ~US\$600-US\$3000) ^{39,40}	Medium (consumables: ~US\$10; control system: ~US\$100-US\$1000) ^{41,43,47}	

Comment 3.4: Please include a discussion on the limitations of EPD. This can include information on the size of droplets that EPD can handle, the characteristics of the surrounding medium, and other different EPD parameters.

Response: We thank the Reviewer for this valuable comment, and we have included the following discussions on the limitations of EPD to the revised manuscript:

1. **Surrounding medium (Line 312-314):** “Droplet in the surroundings of air (Fig. 4G), oil-air interface (Fig. 4H), and oil (Fig. 4I) can all be actuated by EPD, as long as the permittivity of the droplet is different from that of the surrounding medium (Equation 1).”

2. **Hydrophilic substrate (Line 314-317):** “As for the substrate, unlike EWOD^{8,9}, EPD has no requirements for substrate material or thickness, but it still requires a hydrophobic surface (fig. S13) or oil-substrate to reduce the actuation resistance and droplet residue. The actuation on hydrophilic surfaces would be limited by the increasing resistance (fig. S14, note S4).”

3. **Actuation precision (Line 389-391):** “The actuation resolution is determined by the size of the coil, i.e., ~1.5 mm in this case, while the actuation precision is related to the movement of the actuation magnet, slightly lower than that of EWOD but comparable with a magnetic-based droplet actuation platform¹⁷ (fig. S21).”

4. **Droplet size and system downsizing (Line 507-514):** “From the technical perspective, the system could be miniaturized and modularized to achieve fluid manipulation at the nanoscale. The system we show in the paper is mainly designed to manipulate droplets on a μL scale (500 nL-1 mL). If we downsize the system while maintaining the original design, the charge density of the electret needs to be increased to ensure functionality (fig. S28). Besides, we can also microfabricate a 2D charge distribution-controllable electret material. This approach has the potential to actuate droplets by directly programming the local electric field distribution without physically moving the electret material, thus further reducing the size of the system.”

5. **Controllability and speed (Line 514-522):** “Another promising area of technical development lies in the improvement of controllability and speed. At the actuation system level, the magnetic force generated by the programmable control matrix can be further enhanced by increasing the current (e.g. replacing switch ICs with higher current threshold chips) or modifying the coil design (e.g. more layers of coils), increasing system robustness and switching frequency. At the actuation material level, we can utilize different electret materials (e.g. CYTOP, which can provide a higher surface charge density up to 2 mC/m^2 for a $15\text{-}\mu\text{m}$ thick film¹) or apply different charging methods (e.g. corona charging or electron-beam irradiation²) to further increase the surface charge density of the electret, providing a larger actuation force (Note S2).”

Comment 3.5: In the supplementary videos for the gripper platform, it is not clear whether the droplet is moving around the vicinity of any of the four electrets or is precisely stable at its intended position, which is the centre of the four electrets. Please explain.

Response: We thank the Reviewer for raising this issue, which made us realize that our description of EPD gripper is not yet clear enough. According to the analysis of Main Fig. 5G, the forces generated by the EPD gripper have four stable equilibrium points at four different locations. Therefore, droplets nearby will be attracted and stabilized at any one of the equilibrium points. As a result, the intended position of droplet is not the center of the four electrets, but rather any of the equilibrium points around the vicinity of four electrets (newly added Supplementary Fig. S18), consistent with the situation observed in the supplementary videos. To address the comment, we have added the following figure and updated the description in the revised manuscript (Line 373-375).

Figure S18. Locations of the four stable equilibrium points generated by EPD gripper according to the simulation result.

Comment 3.6: Please detail the comparison of resolution and precision of EPD compared to other techniques (like EWOD), including some dimensional data, such as the mean resolution and standard deviation, for their ability to move/displace liquid droplets.

Response: We thank the Reviewer for this valuable comment. We have performed additional experiments to compare the resolution and precision of EPD, EWOD, and magnetic-based system (newly added Supplementary Fig. S21). Resolution is defined as the actuation distance in one step, which depends on the coil/electrode/electromagnet size of the platform. In experiments, the resolutions attainable by the three systems

are essentially comparable. As for the precision, it is negatively correlated with the standard deviation of actuation distance per step. The results indicate that EWOD has the highest precision, followed by the EPD and the magnetic-based system. To address this comment, we have added the following figure and corresponding descriptions in the revised manuscript (Line 389-391).

Figure S21. Resolution and precision of EPD compared to EWOD and magnetic-based system in experiments. Error bars, SD (n = 9). Scale bar: 2 mm.

Comment 3.7: Important aspects related to the coalescence or merging of droplets in an oil medium have been missed. It would be helpful if the authors could provide some explanation about these aspects to aid the readers' comprehension. Authors can use this literature for references: Manipulation and control of droplets on surfaces in a homogeneous electric field | Nature Communications; Effect of electrowetting induced capillary oscillations on coalescence of compound droplets - ScienceDirect.

Response: We are grateful for the inspiring comment from the Reviewer. To address the comment, we have added the above-mentioned literatures as references. We have also included additional figure (newly added Supplementary Fig. S23) and analysis (newly added Supplementary Note S5) related to the merging of droplets in the revised manuscript (Line 393-396): “Although two droplets floating at the oil-air interface can gradually approach and eventually merge under capillary force (Note S5)⁶⁸⁻⁷², the presence of an EPD gripper can further accelerate their approach and merging through attractive forces (fig. S23).”

Figure S23. Analysis of droplet merging in EPD-based droplet robotic system. (A) The lateral optical image of the floating water droplet at the oil-air interface. (B) Schematic diagram of two floating water droplets at the oil-air interface, showing the distorted interface will lead to the attractive force. (C) Simulated EPD force applied on two droplets by EPD gripper, indicating that both droplets are subjected to EPD forces pointing at each other. (D) Time required for two 20 μL droplets merging, indicating the presence of EPD gripper can help to promote droplet merging.

Comment 3.8: Please review the manuscript carefully for any grammatical errors or typos and correct them as needed.

Response: We thank the Reviewer for examining our manuscript. We have carefully reviewed the manuscript and corrected any grammatical errors or typos, as blue-highlighted.

References for point-to-point response:

- 1 Kashiwagi, K. *et al.* Nano-cluster-enhanced high-performance perfluoro-polymer electrets for energy harvesting. *Journal of Micromechanics and Microengineering* **21**, 125016, doi:10.1088/0960-1317/21/12/125016 (2011).
- 2 Suzuki, Y. Recent progress in MEMS electret generator for energy harvesting. *IEEJ Transactions on Electrical and Electronic Engineering* **6**, 101-111, doi:<https://doi.org/10.1002/tee.20631> (2011).
- 3 Chen, J., Yu, Y., Li, J., Lai, Y. & Zhou, J. Size-variable droplet actuation by interdigitated electrowetting electrode. *Applied Physics Letters* **101**, doi:10.1063/1.4769433 (2012).
- 4 Gong, J. & Kim, C.-J. C. All-electronic droplet generation on-chip with real-time feedback control for EWOD digital microfluidics. *Lab on a Chip* **8**, 898-906, doi:10.1039/B717417A (2008).
- 5 Sung Kwon, C., Hyejin, M. & Chang-Jin, K. Creating, transporting, cutting, and merging liquid droplets by electrowetting-based actuation for digital microfluidic circuits. *Journal of Microelectromechanical Systems* **12**, 70-80, doi:10.1109/JMEMS.2002.807467 (2003).
- 6 Fan, S.-K., Hsu, Y.-W. & Chen, C.-H. Encapsulated droplets with metered and removable oil shells by electrowetting and dielectrophoresis. *Lab on a Chip* **11**, 2500-2508, doi:10.1039/C1LC20142E (2011).
- 7 Latip, E. A. *et al.* Protein droplet actuation on superhydrophobic surfaces: a new approach toward anti-biofouling electrowetting systems. *RSC advances* **7**, 49633-49648 (2017).
- 8 Chae, J. B. *et al.* Optimum thickness of hydrophobic layer for operating voltage reduction in EWOD systems. *Sensors and Actuators A: Physical* **215**, 8-16, doi:<https://doi.org/10.1016/j.sna.2013.11.001> (2014).
- 9 Liu, H., Dharmatilake, S., Maurya, D. K. & Tay, A. A. Dielectric materials for electrowetting-on-dielectric actuation. *Microsystem technologies* **16**, 449-460 (2010).
- 10 Brassard, D., Malic, L., Normandin, F., Tabrizian, M. & Veres, T. Water-oil core-shell droplets for electrowetting-based digital microfluidic devices. *Lab on a Chip* **8**, 1342-1349 (2008).
- 11 Shen, H.-H., Fan, S.-K., Kim, C.-J. & Yao, D.-J. EWOD microfluidic systems for biomedical applications. *Microfluidics and Nanofluidics* **16**, 965-987, doi:10.1007/s10404-014-1386-y (2014).
- 12 Long, Z., Shetty, A. M., Solomon, M. J. & Larson, R. G. Fundamentals of magnet-actuated droplet manipulation on an open hydrophobic surface. *Lab on a Chip* **9**, 1567-1575, doi:10.1039/B819818G (2009).
- 13 Zhang, Y. & Nguyen, N.-T. Magnetic digital microfluidics – a review. *Lab on a Chip* **17**, 994-1008, doi:10.1039/C7LC00025A (2017).
- 14 Gao, L. *et al.* Intrinsic peroxidase-like activity of ferromagnetic nanoparticles. *Nature nanotechnology* **2**, 577-583 (2007).
- 15 Wei, H. & Wang, E. Fe₃O₄ magnetic nanoparticles as peroxidase mimetics and their applications in H₂O₂ and glucose detection. *Analytical chemistry* **80**, 2250-2254 (2008).
- 16 Lin, H. *et al.* Ferrobotic swarms enable accessible and adaptable automated viral testing. *Nature* **611**, 570-577, doi:10.1038/s41586-022-05408-3 (2022).
- 17 Yu, W. *et al.* A ferrobotic system for automated microfluidic logistics. *Science Robotics* **5**, eaba4411, doi:10.1126/scirobotics.aba4411 (2020).
- 18 Tekin, H. C. & Gijs, M. A. M. Ultrasensitive protein detection: a case for microfluidic magnetic bead-based assays. *Lab on a Chip* **13**, 4711-4739, doi:10.1039/C3LC50477H (2013).
- 19 Yaman, S., Anil-Inevi, M., Ozcivici, E. & Tekin, H. C. Magnetic Force-Based Microfluidic Techniques for Cellular and Tissue Bioengineering. *Frontiers in Bioengineering and Biotechnology* **6**, doi:10.3389/fbioe.2018.00192 (2018).
- 20 Bajwa, I. U. & Sigaud, S. Effect of cell isolation magnetic particles on DNA quantification by UV absorbance spectrophotometry. *medRxiv*, 2023.2004. 2024.23288526 (2023).

21 Seo, K. S., Wi, R., Im, S. G. & Kim, D. H. A superhydrophobic magnetic elastomer actuator for droplet motion control. *Polymers for Advanced Technologies* **24**, 1075-1080, doi:<https://doi.org/10.1002/pat.3190> (2013).

22 Zhou, Q., Ristenpart, W. D. & Stroeve, P. Magnetically Induced Decrease in Droplet Contact Angle on Nanostructured Surfaces. *Langmuir* **27**, 11747-11751, doi:10.1021/la2024633 (2011).

23 Cao, Q., Han, X. & Li, L. Configurations and control of magnetic fields for manipulating magnetic particles in microfluidic applications: magnet systems and manipulation mechanisms. *Lab on a Chip* **14**, 2762-2777, doi:10.1039/C4LC00367E (2014).

24 Guttenberg, Z. *et al.* Planar chip device for PCR and hybridization with surface acoustic wave pump. *Lab on a Chip* **5**, 308-317, doi:10.1039/B412712A (2005).

25 Du, X. Y., Fu, Y. Q., Luo, J. K., Flewitt, A. J. & Milne, W. I. Microfluidic pumps employing surface acoustic waves generated in ZnO thin films. *Journal of Applied Physics* **105**, doi:10.1063/1.3068326 (2009).

26 Wang, Z. & Zhe, J. Recent advances in particle and droplet manipulation for lab-on-a-chip devices based on surface acoustic waves. *Lab on a Chip* **11**, 1280-1285, doi:10.1039/C0LC00527D (2011).

27 Bourquin, Y., Reboud, J., Wilson, R. & Cooper, J. M. Tuneable surface acoustic waves for fluid and particle manipulations on disposable chips. *Lab on a Chip* **10**, 1898-1901, doi:10.1039/C004506C (2010).

28 Chen, X. *et al.* Low-noise fluorescent detection of cardiac troponin I in human serum based on surface acoustic wave separation. *Microsystems & Nanoeengineering* **9**, 141, doi:10.1038/s41378-023-00600-5 (2023).

29 Liu, X. *et al.* Surface acoustic wave based microfluidic devices for biological applications. *Sensors & Diagnostics* **2**, 507-528 (2023).

30 Girardo, S., Cecchini, M., Beltram, F., Cingolani, R. & Pisignano, D. Polydimethylsiloxane–LiNbO₃ surface acoustic wave micropump devices for fluid control into microchannels. *Lab on a Chip* **8**, 1557-1563, doi:10.1039/B803967D (2008).

31 Renaudin, A., Chabot, V., Grondin, E., Aimez, V. & Charette, P. G. Integrated active mixing and biosensing using surface acoustic waves (SAW) and surface plasmon resonance (SPR) on a common substrate. *Lab on a Chip* **10**, 111-115, doi:10.1039/B911953A (2010).

32 Mutafooulos, K., Lu, P. J., Garry, R., Spink, P. & Weitz, D. A. Selective cell encapsulation, lysis, pico-injection and size-controlled droplet generation using traveling surface acoustic waves in a microfluidic device. *Lab on a Chip* **20**, 3914-3921 (2020).

33 Salehi-Reyhani, A. *et al.* Chemical-free lysis and fractionation of cells by use of surface acoustic waves for sensitive protein assays. *Analytical chemistry* **87**, 2161-2169 (2015).

34 Li, H., Friend, J., Yeo, L., Dasvarma, A. & Traianedes, K. Effect of surface acoustic waves on the viability, proliferation and differentiation of primary osteoblast-like cells. *Biomicrofluidics* **3**, doi:10.1063/1.3194282 (2009).

35 Yeo, L. Y. & Friend, J. R. Ultrafast microfluidics using surface acoustic waves. *Biomicrofluidics* **3**, doi:10.1063/1.3056040 (2009).

36 Renaudin, A., Tabourier, P., Zhang, V., Camart, J. C. & Druon, C. SAW nanopump for handling droplets in view of biological applications. *Sensors and Actuators B: Chemical* **113**, 389-397, doi:<https://doi.org/10.1016/j.snb.2005.03.100> (2006).

37 Cecchini, M., Girardo, S., Pisignano, D., Cingolani, R. & Beltram, F. Acoustic-counterflow microfluidics by surface acoustic waves. *Applied Physics Letters* **92**, doi:10.1063/1.2889951 (2008).

38 Ai, Y. & Marrone, B. L. Droplet translocation by focused surface acoustic waves. *Microfluidics and Nanofluidics* **13**, 715-722, doi:10.1007/s10404-012-0990-y (2012).

39 Ding, X. *et al.* On-chip manipulation of single microparticles, cells, and organisms using surface acoustic waves. *Proceedings of the National Academy of Sciences* **109**, 11105-11109, doi:doi:10.1073/pnas.1209288109 (2012).

40 Zhou, Q., Sariola, V., Latifi, K. & Liimatainen, V. Controlling the motion of multiple objects on a Chladni plate. *Nature Communications* **7**, 12764, doi:10.1038/ncomms12764 (2016).

41 Darhuber, A. A., Valentino, J. P. & Troian, S. M. Planar digital nanoliter dispensing system based on thermocapillary actuation. *Lab on a Chip*

- 10, 1061-1071, doi:10.1039/B921759B (2010).
- 42 Nguyen, N.-T. & Huang, X. Thermocapillary Effect of a Liquid Plug in Transient Temperature Fields. *Japanese Journal of Applied Physics* **44**, 1139, doi:10.1143/JJAP.44.1139 (2005).
- 43 Darhuber, A. A., Valentino, J. P., Troian, S. M. & Wagner, S. Thermocapillary actuation of droplets on chemically patterned surfaces by programmable microheater arrays. *Journal of Microelectromechanical Systems* **12**, 873-879, doi:10.1109/JMEMS.2003.820267 (2003).
- 44 Meng, J. *et al.* AC electrothermal mixing for high conductive biofluids by arc-electrodes. *Journal of Micromechanics and Microengineering* **28**, 065004, doi:10.1088/1361-6439/aab39b (2018).
- 45 Won, B. J., Lee, W. & Song, S. Estimation of the thermocapillary force and its applications to precise droplet control on a microfluidic chip. *Sci Rep* **7**, 3062, doi:10.1038/s41598-017-03028-w (2017).
- 46 Sammarco, T. S. & Burns, M. A. Thermocapillary pumping of discrete drops in microfabricated analysis devices. *AIChE Journal* **45**, 350-366, doi:<https://doi.org/10.1002/aic.690450215> (1999).
- 47 Liu, M.-C. *et al.* Two dimensional thermoelectric platforms for thermocapillary droplet actuation. *RSC Advances* **2**, 1639-1642, doi:10.1039/C1RA00896J (2012).
- 48 Prakash, R., Papageorgiou, D. P., Papathanasiou, A. G. & Kaler, K. V. I. S. Dielectrophoretic liquid actuation on nano-textured super hydrophobic surfaces. *Sensors and Actuators B: Chemical* **182**, 351-361, doi:<https://doi.org/10.1016/j.snb.2013.03.024> (2013).
- 49 Dai, H. *et al.* Controllable High-Speed Electrostatic Manipulation of Water Droplets on a Superhydrophobic Surface. *Advanced Materials* **31**, 1905449, doi:<https://doi.org/10.1002/adma.201905449> (2019).
- 50 Jin, Y. *et al.* Charge-Powered Electrotaxis for Versatile Droplet Manipulation. *ACS Nano* **17**, 10713-10720, doi:10.1021/acsnano.3c01919 (2023).
- 51 Zheng, L. *et al.* Self-Powered Electrostatic Actuation Systems for Manipulating the Movement of both Microfluid and Solid Objects by Using Triboelectric Nanogenerator. *Advanced Functional Materials* **27**, 1606408, doi:<https://doi.org/10.1002/adfm.201606408> (2017).
- 52 Jin, Y. *et al.* Electrostatic tweezer for droplet manipulation. *Proceedings of the National Academy of Sciences* **119**, e2105459119, doi:10.1073/pnas.2105459119 (2022).
- 53 Pethig, R. Review Article—Dielectrophoresis: Status of the theory, technology, and applications. *Biomicrofluidics* **4**, doi:10.1063/1.3456626 (2010).
- 54 Çetin, B. & Li, D. Dielectrophoresis in microfluidics technology. *ELECTROPHORESIS* **32**, 2410-2427, doi:<https://doi.org/10.1002/elps.201100167> (2011).
- 55 Renaudot, R. *et al.* Optimization of Liquid DiElectroPhoresis (LDEP) Digital Microfluidic Transduction for Biomedical Applications. *Micromachines* **2**, 258-273 (2011).
- 56 Fan, S.-K., Hsieh, T.-H. & Lin, D.-Y. General digital microfluidic platform manipulating dielectric and conductive droplets by dielectrophoresis and electrowetting. *Lab on a Chip* **9**, 1236-1242, doi:10.1039/B816535A (2009).
- 57 Velev, O. D., Prevo, B. G. & Bhatt, K. H. On-chip manipulation of free droplets. *Nature* **426**, 515-516, doi:10.1038/426515a (2003).
- 58 Zhao, K. & Li, D. Direct current dielectrophoretic manipulation of the ionic liquid droplets in water. *Journal of Chromatography A* **1558**, 96-106, doi:<https://doi.org/10.1016/j.chroma.2018.05.020> (2018).
- 59 Voldman, J. Electrical forces for microscale cell manipulation. *Annual Review of Biomedical Engineering* **8**, 425-454, doi:10.1146/annurev.bioeng.8.061505.095739 (2006).
- 60 Hakoda, M. & Hirota, Y. Correlation between dielectric property by dielectrophoretic levitation and growth activity of cells exposed to electric field. *Bioprocess and Biosystems Engineering* **36**, 1219-1227, doi:10.1007/s00449-012-0849-3 (2013).
- 61 Kaler, K. V. I. S., Prakash, R. & Chugh, D. Liquid dielectrophoresis and surface microfluidics. *Biomicrofluidics* **4**, doi:10.1063/1.3411003

(2010).

- 62 Kwak, T. J., Hossen, I., Bashir, R., Chang, W.-J. & Lee, C. H. Localized Dielectric Loss Heating in Dielectrophoresis Devices. *Sci Rep* **9**, 18977, doi:10.1038/s41598-019-55031-y (2019).
- 63 Greene, C. *et al.* Blood–brain barrier disruption and sustained systemic inflammation in individuals with long COVID-associated cognitive impairment. *Nature Neuroscience* **27**, 421–432, doi:10.1038/s41593-024-01576-9 (2024).
- 64 Jones, M. A., Töttemeyer, S., Maskell, D. J., Bryant, C. E. & Barrow, P. A. Induction of Proinflammatory Responses in the Human Monocytic Cell Line THP-1 by *Campylobacter jejuni*. *Infection and Immunity* **71**, 2626–2633, doi:10.1128/iai.71.5.2626-2633.2003 (2003).
- 65 Broderick, L. & Hoffman, H. M. IL-1 and autoinflammatory disease: biology, pathogenesis and therapeutic targeting. *Nature Reviews Rheumatology* **18**, 448–463, doi:10.1038/s41584-022-00797-1 (2022).
- 66 Nature Communications. *Aims & scope*, <<https://www.nature.com/ncomms/aims>> (2024).
- 67 Srinivasan, V., Pamula, V. K., Pollack, M. G. & Fair, R. B. in *Proc. MicroTAS*. 1287–1290.
- 68 Bansal, S. & Sen, P. Effect of electrowetting induced capillary oscillations on coalescence of compound droplets. *Journal of Colloid and Interface Science* **530**, 223–232, doi:<https://doi.org/10.1016/j.jcis.2018.05.090> (2018).
- 69 Wang, T., Andersen, S. I. & Shapiro, A. Coalescence of oil droplets in microchannels under brine flow. *Colloids and Surfaces A: Physicochemical and Engineering Aspects* **598**, 124864, doi:<https://doi.org/10.1016/j.colsurfa.2020.124864> (2020).
- 70 Aarts, D. G. A. L., Lekkerkerker, H. N. W., Guo, H., Wegdam, G. H. & Bonn, D. Hydrodynamics of Droplet Coalescence. *Physical Review Letters* **95**, 164503, doi:10.1103/PhysRevLett.95.164503 (2005).
- 71 Pitois, O., Moucheront, P. & Chateau, X. Liquid Bridge between Two Moving Spheres: An Experimental Study of Viscosity Effects. *Journal of Colloid and Interface Science* **231**, 26–31, doi:<https://doi.org/10.1006/jcis.2000.7096> (2000).
- 72 Hartmann, J., Schür, M. T. & Hardt, S. Manipulation and control of droplets on surfaces in a homogeneous electric field. *Nature Communications* **13**, 289, doi:10.1038/s41467-021-27879-0 (2022).

REVIEWERS' COMMENTS

Reviewer #1 (Remarks to the Author):

The comments from the reviewer have been satisfactorily addressed. I think the manuscript is ready for publication.

Reviewer #2 (Remarks to the Author):

The authors have now comprehensively revised the manuscript based on the comments provided by the referees. However, I still believe that the manuscript lacks novelty to fit into the scope of Nature Communications. Thus, I cannot recommend for publication here. However, I encourage the authors to submit the manuscript in the more specialized journals.

Reviewer #3 (Remarks to the Author):

All reviewer comments have been addressed satisfactorily, and I find the paper to be publishable as is.

Response to the reviewed manuscript:

“A Droplet Robotic System Enabled by Electret-induced Polarization on Droplet”

Reviewer #1

The comments from the reviewer have been satisfactorily addressed. I think the manuscript is ready for publication.

Response: We appreciate the reviewer for all suggestions raised, which have greatly improved the quality of our manuscript.

Reviewer #2

The authors have now comprehensively revised the manuscript based on the comments provided by the referees. However, I still believe that the manuscript lacks novelty to fit into the scope of Nature Communications. Thus, I cannot recommend for publication here. However, I encourage the authors to submit the manuscript in the more specialized journals.

Response: We would like to thank the reviewer for all the constructive comments, which have helped us to make great improvements to our manuscript. The comments have inspired us to emphasize the scientific novelty and quantitative achievement of this paper, and systematically compare the differences between our work and other existing techniques. The novelty of our work in both mechanism and application aspects is thus further explored and demonstrated, including the unique EPD mechanism, superior adaptability with various liquids/biochemical samples, the design of a fully automated system and its implication on multidisciplinary liquid-based applications. These have now been included in the revised manuscript and we sincerely appreciate the reviewers' effort in evaluating our work and proposing all the valuable suggestions.

Reviewer #3

All reviewer comments have been addressed satisfactorily, and I find the paper to be publishable as is.

Response: We are very thankful to the reviewer's helpful and high-quality suggestions in improving our manuscript.